# Peroxisome biogenesis deficiency attenuates the BDNF-TrkB pathway-mediated development of the cerebellum

Yuichi Abe[1], Masanori Honsho[1], Ryota Itoh[2], Ryoko Kawaguchi[2], Masashi Fujitani[3], Kazushirou Fujiwara[2], Masaaki Hirokane[2], Takashi Matsuzaki[2], Keiko Nakayama[4,5], Ryohei Ohgi[2], Toshihiro Marutani[2], Keiichi I Nakayama[4], Toshihide Yamashita[3,6], Yukio Fujiki[1]

Peroxisome biogenesis disorders (PBDs) manifest as neurological deficits in the central nervous system, including neuronal migration defects and abnormal cerebellum development. However, the mechanisms underlying pathogenesis remain enigmatic. Here, to investigate how peroxisome deficiency causes neurological defects of PBDs, we established a new PBD model mouse defective in peroxisome assembly factor Pex14p, termed *Pex14^(ΔC/ΔC)* mouse. *Pex14^(ΔC/ΔC)* mouse manifests a severe symptom such as disorganization of cortical laminar structure and dies shortly after birth, although peroxisomal biogenesis and metabolism are partially defective. The *Pex14^(ΔC/ΔC)* mouse also shows malformation of the cerebellum including the impaired dendritic development of Purkinje cells. Moreover, extracellular signal-regulated kinase and AKT signaling are attenuated in this mutant mouse by an elevated level of brain-derived neurotrophic factor (BDNF) together with the enhanced expression of TrkB-T1, a dominant-negative isoform of the BDNF receptor. Our results suggest that dysregulation of the BDNF-TrkB pathway, an essential signaling for cerebellar morphogenesis, gives rise to the pathogenesis of the cerebellum in PBDs.

## Introduction

The peroxisome serves as a platform for various catabolic and anabolic reactions, such as β-oxidation of very long–chain fatty acids (VLCFAs), degradation of hydrogen peroxide, and plasmalogen biogenesis (Wanders & Waterham, 2006). The physiological consequence of peroxisomal function is highlighted by the pathogenesis of peroxisome biogenesis disorders (PBDs), autosomal recessive diseases manifesting as progressive disorders of the central nervous system (CNS) (Weller et al, 2003; Steinberg et al, 2006). PBDs, including Zellweger spectrum disorders (ZSDs), rhizomelic chondrodysplasia punctata type 1 (RCDP1) (Braverman et al, 1997; Motley et al, 1997; Purdue et al, 1997), and RCDP5 (Barøy et al, 2015), are caused by mutations of *PEX* genes encoding peroxins required for peroxisome assembly (Waterham & Ebberink, 2012; Fujiki et al, 2014; Fujiki, 2016). The primary defects of RCDP1 and RCDP5 are the loss of *PEX7* and the long isoform of *PEX5*, respectively, whereas mutations in any of the other *PEX* genes give rise to the ZSD. ZSDs, accounting for about 80% of the PBD patients (Weller et al, 2003), are classified into three groups according to their clinical severity: Zellweger syndrome (ZS), neonatal adrenoleukodystrophy (NALD), and infantile Refsum disease (IRD) (Steinberg et al, 2006). Patients with ZS, the most severe ZSDs, generally die before reaching the age of 1 yr. The CNS pathological features of patients with ZS include migration defects in cortical neurons, abnormal dendritic arborization of Purkinje cells, and dysplastic alterations of inferior olivary nuclei (ION) (Volpe & Adams, 1972; de León et al, 1977; Evrard et al, 1978; Steinberg et al, 2006). The biochemical abnormalities, including marked reduction of plasmalogens, accumulation of VLCFAs, and reduction in the level of docosahexaenoic acid (DHA) (Weller et al, 2003), are thought to be relevant to the manifestations of malformations in the CNS. However, the pathogenic mechanisms of PBDs are largely unknown.

To study the pathogenesis of ZSDs, mice with generalized inactivation of the *Pex* genes *Pex2*, *Pex5*, and *Pex13* have been established (Baes et al, 1997; Faust & Hatten, 1997; Maxwell et al, 2003). The deletion of individual *Pex* genes causes the complete deficiency of peroxisomal protein import and abnormal

[1]Division of Organelle Homeostasis, Medical Institute of Bioregulation, Kyushu University, Fukuoka, Japan    [2]Graduate School of Systems Life Sciences and Department of Biology, Faculty of Sciences, Kyushu University Graduate School, Fukuoka, Japan    [3]Department of Molecular Neuroscience, Graduate School of Medicine, Osaka University, Osaka, Japan    [4]Department of Molecular and Cellular Biology, Medical Institute of Bioregulation, Kyushu University, Fukuoka, Japan    [5]Division of Cell Proliferation, Tohoku University Graduate School of Medicine, Sendai, Japan    [6]Core Research for Evolutional Science and Technology, Japan Science and Technology Agency, Tokyo, Japan

Correspondence: yfujiki@kyudai.jp

morphology of the CNS (Baes et al, 1997; Faust & Hatten, 1997; Faust, 2003; Maxwell et al, 2003), as reported in patients with ZS (Volpe & Adams, 1972; Evrard et al, 1978; Powers & Moser, 1998). Moreover, the mutation of *Pex* genes in the CNS results in dysfunction of peroxisomes in neurons, oligodendrocytes, and astrocytes, giving rise to abnormal development and aberrant brain morphology (Krysko et al, 2007; Müller et al, 2011), as observed in *Pex*-null mice (Baes et al, 1997; Faust & Hatten, 1997; Faust, 2003; Maxwell et al, 2003). However, mice with neural cell type–selective conditional knockout of *Pex* genes do not show abnormal CNS development (Kassmann et al, 2007; Bottelbergs et al, 2010). Normal development in these mice has been suggested to be due to the shuttling of peroxisomal metabolites and supportive effects among different brain cell types (Bottelbergs et al, 2010). Therefore, investigation of cell–cell interaction between neuronal cells might serve as a potential clue to reveal the pathological mechanisms underlying the abnormal development of neuronal cells. In the present study, as a step toward uncovering pathological mechanisms underlying ZSDs, we established a new ZSD model mouse, defective in *Pex14*. The *Pex14*-defective mouse manifests severe symptoms in CNS and growth retardations, while the peroxisome biogenesis and metabolism are partially defective. Moreover, the up-regulation of brain-derived neurotrophic factor (BDNF) was observed in the cerebellum of a *Pex14*-defective mouse, manifesting the dysmorphogenesis of Purkinje cells. Taken together, our results suggest for the first time the pathogenesis of abnormal cerebellar development in ZSDs.

# Results

### Generation of a *Pex14*-defective mouse

To investigate how peroxisome deficiency causes malformation of CNS in patients with ZSDs, we established a *Pex14* mutant mouse with deletion of the C-terminal half part of Pex14p by eliminating exons 6–8 from the *Pex14* gene on a C57BL/6 background, termed *Pex14^{ΔC/ΔC}* mouse (Fig 1A and B). This deletion of exons 6–8 induced a frameshift of the amino acid at position 129 and generated premature termination at position 164 (Fig 1C, middle), giving rise to the C-terminal–truncated mutant of Pex14p similar to that found in a patient with ZS (Shimozawa et al, 2004) (Pex14p-Q185X, Fig 1C, bottom). The patient with Pex14p-Q185X mutation manifested severe CNS defects, such as hypotonia and psychomotor retardation, and died at the age of 10 d (Shimozawa et al, 2004). Nevertheless, skin fibroblasts from the patient showed partial defects in peroxisomal biogenesis and metabolism (Fig S1).

*Pex14^{ΔC/ΔC}* pups were generated by breeding heterozygous *Pex14* mutants, and the expected Mendelian ratio of 22 wild-type mice (*Pex14^{+/+}*, 24%), 48 heterozygous mice (*Pex14^{+/ΔC}*, 53%), and 21 homozygous mutant mice (*Pex14^{ΔC/ΔC}*, 23%) was obtained (Fig 1E). However, homozygous neonates died shortly after birth or within several days (data not shown). Neonates with the mutation were smaller in body size than those of wild-type mice at birth (Fig 1D), and body weights of neonatal *Pex14^{ΔC/ΔC}* mice were indeed lower at P0.5 (Fig 1E). Cresyl violet staining of coronal sections revealed that

neurons migrating to the cortex accumulated in the intermediate zone at the medial region (Fig 1F) as previously reported in other *Pex*-knockout mice (Baes et al, 1997; Faust & Hatten, 1997; Janssen et al, 2003; Maxwell et al, 2003). In the cortical layer, cells with elongated shapes such as pyramidal cells are normally localized in cortical layer V of the wild-type mouse (Fig 1G, left panel). By contrast, in the brain of the *Pex14^{ΔC/ΔC}* mouse, more darkly stained cells with rounder shapes were accumulated in layer V and the boundary was obscured between layers IV and V (Fig 1G, right panel).

Next, we analyzed peroxisomal biogenesis in the *Pex14^{ΔC/ΔC}* mouse brain. Expression and peroxisomal localization of Pex14p truncated at the C-terminal portion were detected with an antibody to the N-terminal region of Pex14p (Pex14pN) (Itoh & Fujiki, 2006) (Fig 2D and E), but not with an antibody against the C-terminal portion of Pex14p (Pex14pC) in *Pex14^{ΔC/ΔC}* mouse brain (Fig 2A, B, D, and E). Catalase was diffused throughout the cytosol (Fig 2C) and import of peroxisomal targeting signal 1 (PTS1) and PTS2 proteins was partially defective in the *Pex14^{ΔC/ΔC}* mouse (Fig 2A, B, and E) as shown by the reduced level of intraperoxisomal conversion of the acyl-CoA oxidase 1 (AOx) A-chain to B-chain (Fig 2E) and C-chain (not shown) (Miyazawa et al, 1989) and the fluorescence intensity for alkyldihydroxyacetonephosphate synthase (ADAPS) (Fig 2B). Peroxisomal membranes were discernible by staining with an antibody to PMP70 and appeared similar to those observed in the wild-type mouse brain (Fig 2C). Taken together, these results indicated that the C-terminal deletion of Pex14p gave rise to a partial defect in peroxisomal matrix protein import and wholly impaired catalase import.

We examined phospholipid metabolism in the mouse brain by liquid chromatography coupled with tandem mass spectrometry (LC-MS/MS) analysis. The plasmenylethanolamine (PlsEtn) level in the *Pex14^{ΔC/ΔC}* mouse brain was reduced to nearly 50% of that in the wild-type mouse (Fig 2F). Typical metabolic abnormalities in peroxisome biogenesis deficiency, including accumulation of VLCFA-containing phosphatidylcholine (VLCPC; Fig 2G) and decrease in DHA-containing phospholipids (DHA-PLs; Fig 2H), were also evident in the *Pex14^{ΔC/ΔC}* mouse brain. However, these metabolic aberrations in *Pex14^{ΔC/ΔC}* mice were milder than those in other *Pex*-knockout mice, such as 100-fold reduction of PlsEtn and sixfold or more accumulation of VLCFAs (Baes et al, 1997; Faust et al, 2001; Maxwell et al, 2003). Therefore, partial defect of peroxisomal protein import causes the mild metabolic abnormalities in the *Pex14^{ΔC/ΔC}* mice. The heterozygous *Pex14^{+/ΔC}* mouse showed no anomalies in growth (Fig 1D and E), AOx conversion (Fig 2E), and peroxisomal lipid metabolism (Fig 2F–H), whereas Pex14p was markedly reduced (Fig 2E).

Collectively, these data demonstrated that mild defects in peroxisome biogenesis and metabolism are sufficient to manifest as the symptoms of ZS. Thus, we concluded that the *Pex14^{ΔC/ΔC}* mouse was a good model of ZS. However, morphological abnormalities of the CNS other than migration defect of cortical neurons (Fig 1F and G and S2), such as cerebellar malformation in the patients with ZS (Volpe & Adams, 1972; de León et al, 1977; Evrard et al, 1978), were not evident in *Pex14^{ΔC/ΔC}* mouse (data not shown). Because the cerebellum undergoes dramatic developmental change during the first three postnatal weeks in mice, it was difficult to verify the dysmorphology of the cerebellum in neonatal *Pex14^{ΔC/ΔC}* mice, which died shortly after birth.

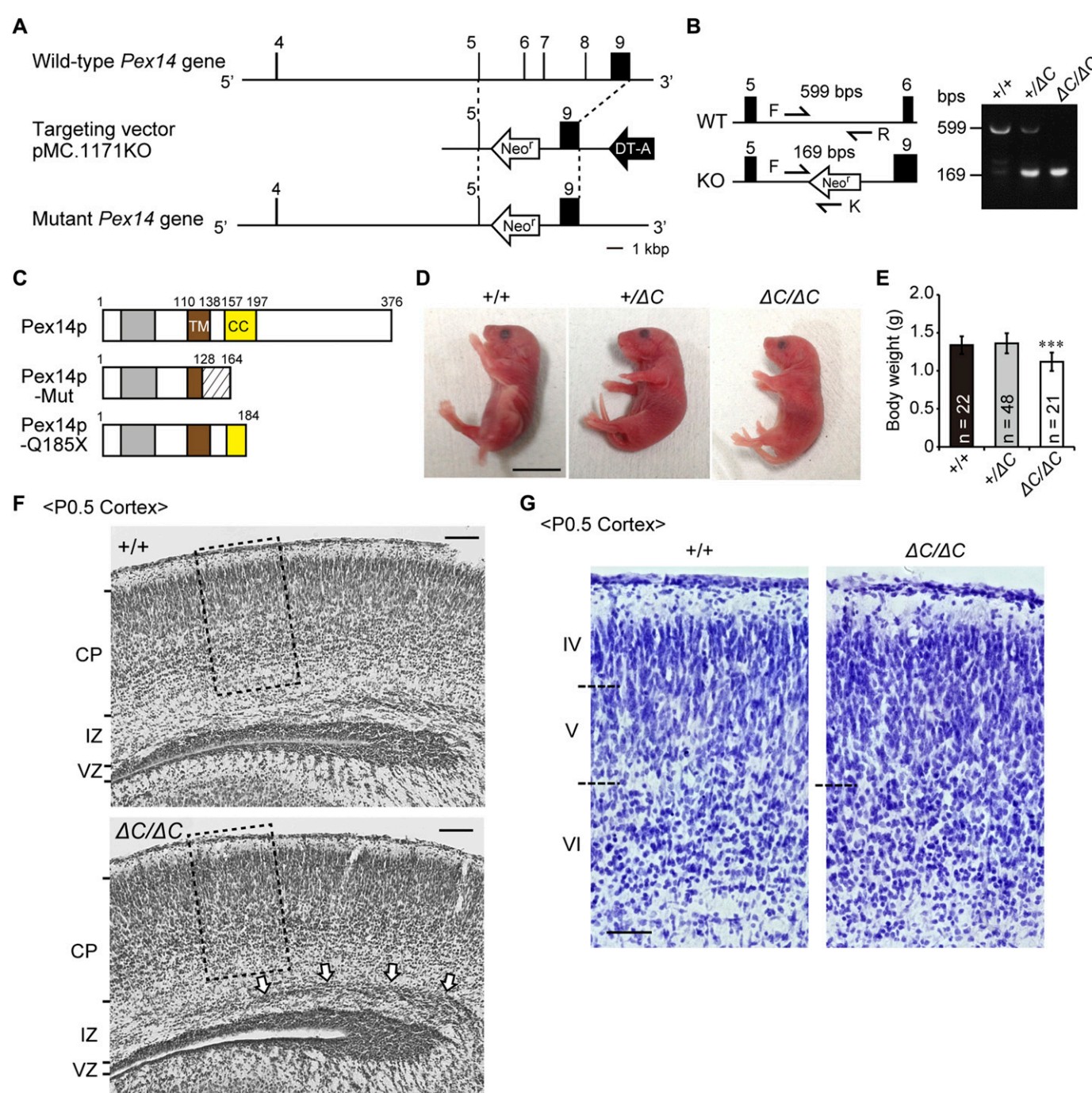

**Figure 1. Targeted disruption of the mouse *Pex14* gene.**
**(A)** Schematic representation of the *Pex14* genome locus (top), targeting vector (pMC-KO, middle), and targeted allele of the mutated locus following the homologous recombination (bottom). Exon sequences are indicated by black bars and boxes. **(B)** PCR-based genotyping using tail-derived DNA of wild-type (+/+), heterozygous (+/ΔC), and homozygous (ΔC/ΔC) *Pex14* mutant mice. Arrows indicate the sequences used for primers described in the Materials and Methods section. Primers P14F (F) and P14R (R) amplify a 599-bp fragment, and primers F and KN52-2 (K) amplify a 169-bp fragment specific for the recombined *Pex14* gene. **(C)** Schematic structure of predicted Pex14p proteins in wild-type mice, *Pex14* mutant mice (Pex14p-Mut), and patients with a *Pex14* nonsense mutation, C553T (Pex14p-Q185X). Gray bar, Pex5p-binding domain; brown bar, transmembrane domain (TM); yellow bar, coiled-coil domain (CC); shaded area, altered amino acid sequence caused by a frameshift mutation (129–163). **(D)** Phenotypic appearance of *Pex14* mutant mice ~12 h after birth. Scale bar, 1 cm. **(E)** Postnatal body weights were determined at P0.5. The number of pups with each genotype is indicated. ***$P < 0.001$, by Dunnett's test compared with +/+. **(F)** Cresyl violet staining of a coronal section of the cortex from a P0.5 mouse. The open arrows indicate accumulated neurons in the intermediate zone (IZ). Scale bar, 100 μm. **(G)** Enlarged view of the boxed regions in F. Cortical layers are indicated on the left. The boundaries of the cortical plates are indicated by dashed lines. Scale bar, 50 μm. CP, cortical plate; DT-A, diphtheria toxin A cassette; Neo^r, Neomycin-resistant gene; VZ, ventricular zone.

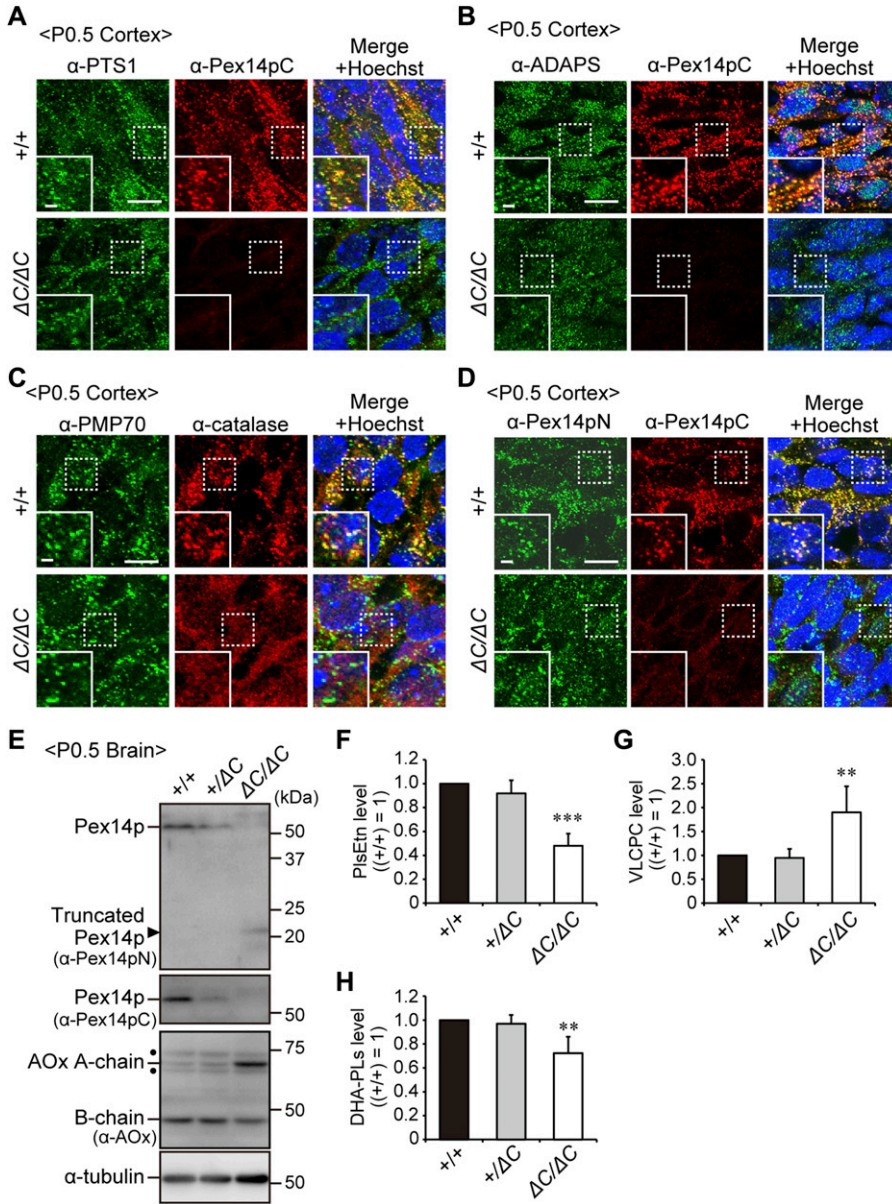

**Figure 2. Impaired peroxisomal biogenesis in the cortex of Pex14$^{ΔC/ΔC}$ mice.**
Immunofluorescence labeling of the cortex in wild-type (+/+) and Pex14$^{ΔC/ΔC}$ (ΔC/ΔC) mice. **(A–D)** The coronal sections of brains were stained with antibodies to PTS1 (green) and Pex14pC (red) (A), ADAPS (green) and Pex14pC (red) (B), PMP70 (green) and catalase (red) (C), or Pex14pN (green) and Pex14pC (red) (D). Staining with Hoechst 33242 (blue) and the merged view are also shown. Scale bar, 10 μm. Higher magnification images of the boxed regions are shown (insets). Scale bar, 2 μm. **(E)** Brain lysates prepared from wild-type, heterozygous (+/ΔC), and homozygous Pex14 mutant mice were subjected to SDS–PAGE and immunoblotting analysis using antibodies against Pex14pN, Pex14pC, AOx, and α-tubulin. Arrowhead indicates C-terminally truncated Pex14p corresponding to Pex14p-Mut. AOx is synthesized as a 75-kD A-chain and converted to a 53-kD B-chain and a 22-kD C-chain in peroxisomes. AOx A and AOx B chains are shown in immunoblots. Dots, non-specific bands. **(F–H)** Total amounts of PlsEtn (F), VLCPC (G), and DHA-PLs (H) are represented relative to those in the wild-type mouse brain (n = 3). **$P < 0.01$, ***$P < 0.001$, by Dunnett's test compared with +/+.
Source data are available for this figure.

## Prolongation of lifespan by replacing the genetic background

Faust (2003) reported that replacement of the Pex2-null allele in a C57BL/6 × 129 Sv background to Swiss Webster × 129SvEv genetic background prolonged lifespan. Thus, a Pex14$^{+/ΔC}$ mouse on a C57BL/6 background was mated with a wild-type mouse on an ICR Swiss background, termed the C57BL/6 × ICR (BL/ICR) strain. Breeding of heterozygous BL/ICR mice generated homozygous mutant BL/ICR newborn mice, termed Pex14$^{ΔC/ΔC}$ BL/ICR mouse, ~30% of which survived over P7 but rarely survived to P14 (Fig 3A). Pups of Pex14$^{ΔC/ΔC}$ BL/ICR mice exhibited severe growth retardation (Fig 3B), regardless of normal suckling activity for several days after the birth, suggesting that the primary cause for the neonatal death is not likely the feeding. This prolongation of the lifespan enabled us to analyze the cerebellar morphologies in Pex14$^{ΔC/ΔC}$ BL/ICR mice.

## Malformation of the cerebellum in Pex14$^{ΔC/ΔC}$ BL/ICR mice

Pex14$^{ΔC/ΔC}$ BL/ICR mice displayed the defects of cerebellar development, including atrophy of the cerebellar folia (Fig 3C), the migration delay of granule cells as suggested by thickening of the external granular layer (Fig 3D and E) and malformation of Purkinje cells at P3 (Fig 3F and G) and P7 (Fig 3I). In wild-type mice at P3, Purkinje cells began to polarize and show elongated cell soma and major dendrites (Fig 3F), and their axons projected into the internal granular layer (IGL, Fig 3G, left panel, arrows). By contrast, Purkinje cells at P3 in Pex14$^{ΔC/ΔC}$ BL/ICR mice were less polarized, the soma was smaller (Fig 3F), and axonal swelling was evident (Fig 3G and H, arrowheads). At P7, impairment of Purkinje cell arborization was more evident and axonal reticular structures were observed in the IGL region (Fig 3I). Such dysmorphogenesis

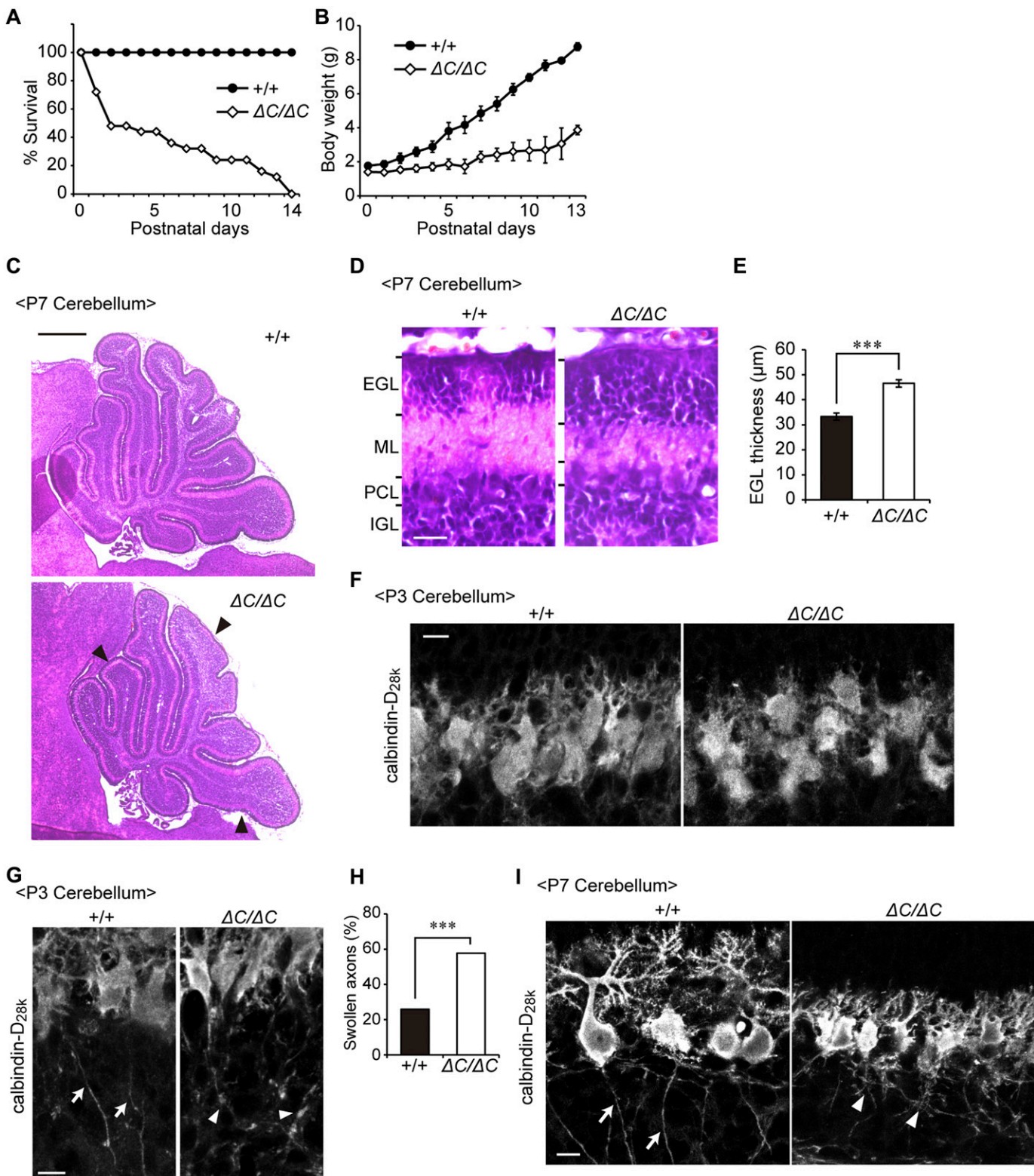

**Figure 3. Defect of cerebellar development and malformation of Purkinje cells in *Pex14^{ΔC/ΔC}* BL/ICR mice.**
**(A)** Percentage of pups surviving at postnatal days. The survival days were based on the pups of 22 wild-type (+/+) and 25 *Pex14^{ΔC/ΔC}* (ΔC/ΔC) BL/ICR mice. **(B)** Body weights of pups at postnatal days were plotted. **(C)** Hematoxylin and eosin staining of the sagittal sections of the cerebellum (P7). Arrowheads indicate the shallow cerebellar folia in the cerebellum of the *Pex14^{ΔC/ΔC}* BL/ICR mouse (lower panel). Scale bar, 500 μm. **(D)** The sagittal section of the cerebellum at P7 was stained with hematoxylin and eosin. Scale bar, 20 μm. **(E)** Thickness of EGL was quantified (n = 3). **(F, G)** Confocal microscopy images of the sagittal sections of the

of Purkinje cells in the postnatal period indicated the defect of cerebellar development, consistent with the results of earlier studies examining *Pex2*- and *Pex13*-knockout mice (Faust, 2003; Müller et al, 2011).

## Peroxisome biogenesis deficiency elevates BDNF expression in neuroblastoma cells

We next examined whether peroxisome biogenesis deficiency alters the neuronal morphology using a neuroblastoma cell line, SH-SY5Y. Knockdown of *PEX5* encoding, an essential cytosolic receptor that binds to Pex14p for import of matrix proteins (Fig 4A), impaired peroxisome biogenesis as indicated by the reduced localization of catalase in peroxisomes (Fig 4B) and induced the cell dispersion and the enhancement of neurite outgrowth in SH-SY5Y (Fig 4C, upper panels). This morphological alteration of SH-SY5Y resembled the BDNF-induced neurite outgrowth on the SH-SY5Y cells (Kaplan et al, 1993). Therefore, we focused on neurotrophins, including NGF, BDNF, neurotrophin 3 (NT-3), and NT-4 (Reichardt, 2006). The neurite outgrowth in SH-SY5Y cells transfected with siRNA against *PEX5* was suppressed by inhibiting the binding of these neurotrophins to their receptors in the presence of recombinant extracellular domain of the p75 neurotrophic receptor (p75ECD-His), common receptor for the neurotrophins (Fig 4C, lower panels) (Reichardt, 2006), suggesting the involvement of neurotrophins in the neurite outgrowth. Real-time PCR revealed that knockdown of *PEX5* elevated *BDNF* expression but not other neurotrophins, including *NGF*, *NT-3*, and *NT-4* (Fig 4D). The intracellular level of BDNF was also increased in *PEX5*-depleted cells (Fig 4E). Moreover, the neurite outgrowth was promoted upon culturing cells in the conditioned medium from *PEX5*-depleted SH-SY5Y cells (Fig 4F, upper panels) and medium containing recombinant BDNF (rBDNF, Fig 4G). However, such morphological changes were suppressed in the presence of p75ECD-His (Fig 4F and G). Taken together, these results suggested that elevated secretion of BDNF from the cells impaired peroxisome biogenesis induced neurite outgrowth in neuronal cells and prompted us to examine the effect of elevated BDNF on neuronal development in the mice defective in peroxisome biogenesis.

## Excess BDNF impairs development of dendrites in *Pex14*-deficient Purkinje cells

To investigate whether the malformation of the cerebellum in *Pex14*$^{\Delta C/\Delta C}$ BL/ICR mice is a consequence of the elevated BDNF protein level, we labeled BDNF in sagittal sections of the cerebellum in P3 mice. BDNF was detected around the Purkinje cell layer in the wild-type mouse, as previously reported (Friedman et al, 1998), and elevated in *Pex14*$^{\Delta C/\Delta C}$ BL/ICR mice (Fig 5A and B). To assess the effect of excess BDNF on the development of Purkinje cells in

*Pex14*$^{\Delta C/\Delta C}$ BL/ICR mouse, we performed primary culture of cerebellar cells. Primary cerebellar neurons isolated from neonatal mouse cerebella (P0.5) were cultured for 14 d in vitro (DIV) in the absence or presence of rBDNF at 50 ng/ml, which is in a range similar to that estimated from concentration of BDNF in rat cerebellum (~10 ng/ml) (Baranowska-Bosiacka et al, 2013). Cerebellar neurons were fixed and stained with anti-calibindin-D$_{28k}$ antibody to visualize Purkinje cells (Fig 5C–H). The axonal elongations and collateral formations were not apparently induced by the treatment with BDNF (Fig 5C–E). Rather, Purkinje cells with swollen axons (Fig 5C, inset) were more frequently detected upon treatment with BDNF of the cells from *Pex14*$^{\Delta C/\Delta C}$ BL/ICR mouse (Fig 5F). Moreover, the development of dendrites in the *Pex14*-defective Purkinje cells was affected in the presence of BDNF (Fig 5G), as determined by the area of Purkinje cell soma and dendrites (Fig 5H). In the wild-type Purkinje cells, the axonal formation and dendritic arborization were not altered by the BDNF treatment (Fig 5C–H). Such dysmorphologies of Purkinje cells in vitro were consistent with those observed in the cerebella of *Pex14*$^{\Delta C/\Delta C}$ BL/ICR mice (Fig 3F–I). Taken together, these results suggested that peroxisome deficiency in Purkinje cells induces the dysregulation of the BDNF signaling pathway in the presence of an elevated level of BDNF, leading to the axonal swelling and the defect of dendritic arborization.

## BDNF-TrkB signaling pathway is attenuated in the cerebellum of *Pex14*-deficient mice

BDNF regulates cell growth and axonal outgrowth via binding to tropomyosin-related kinase B (TrkB) receptor on the neuronal surface (Reichardt, 2006). Functionally, TrkB-defective mutant mice show significant reduction in dendritic arborization of Purkinje cells (Minichiello & Klein, 1996; Rico et al, 2002). Therefore, the BDNF-TrkB signaling pathway is most likely essential for early postnatal development of the cerebellum, especially in the dendritic branching of Purkinje cells (Minichiello & Klein, 1996; Schwartz et al, 1997; Carter et al, 2002; Rico et al, 2002; Sherrard & Bower, 2002).

We next investigated whether expression of TrkB is affected in the cerebellum of *Pex14*$^{\Delta C/\Delta C}$ BL/ICR mice. Immunofluorescent staining of TrkB revealed that TrkB-labeled punctate structures were detected on the IGL side of Purkinje cell soma in the wild-type cerebellum at P3 (Fig 6A, arrowheads). Vesicular glutamate transporter 2 (vGlut2), a marker for climbing fiber terminals, was detected in the same region of Purkinje cells (Fig 6C), suggesting the localization of TrkB on the climbing fiber–Purkinje cell (CF–PC) synapse (Sherrard et al, 2009). Indeed, TrkB-labeled punctate structures partly coincided with or are located adjacent to vGlut2-positive punctate structures (Fig 6D, arrowheads). At P7, TrkB-positive punctate structures were not readily detectable (Fig S3A) and vGlut2 was mainly localized in the dendrite of the Purkinje cells (Fig S3B), suggesting that climbing fiber terminals shifted to the

---

cerebellum at P3 labeled with an antibody to calbindin-D$_{28k}$, a Purkinje cell marker. Arrows indicate axons of wild-type Purkinje cells and arrowheads indicate swollen axons of *Pex14* mutant Purkinje cells. Scale bar, 10 $\mu$m. **(H)** Percentage of swollen axons was quantified (+/+, n = 54; $\Delta C/\Delta C$, n = 52). **(I)** Confocal microscopy images of sagittal sections of the cerebellum at P7 labeled with an antibody to calbindin-D$_{28k}$ are shown. Arrows indicate axons of wild-type Purkinje cells and arrowheads indicate axonal reticular structures of *Pex14* mutant Purkinje cells. Scale bar, 10 $\mu$m. ***$P < 0.001$, by $t$ test (E) and $\chi^2$ test (H). EGL, external granular layer; IGL, internal granular layer; ML, molecular layer; PCL, Purkinje cell layer.

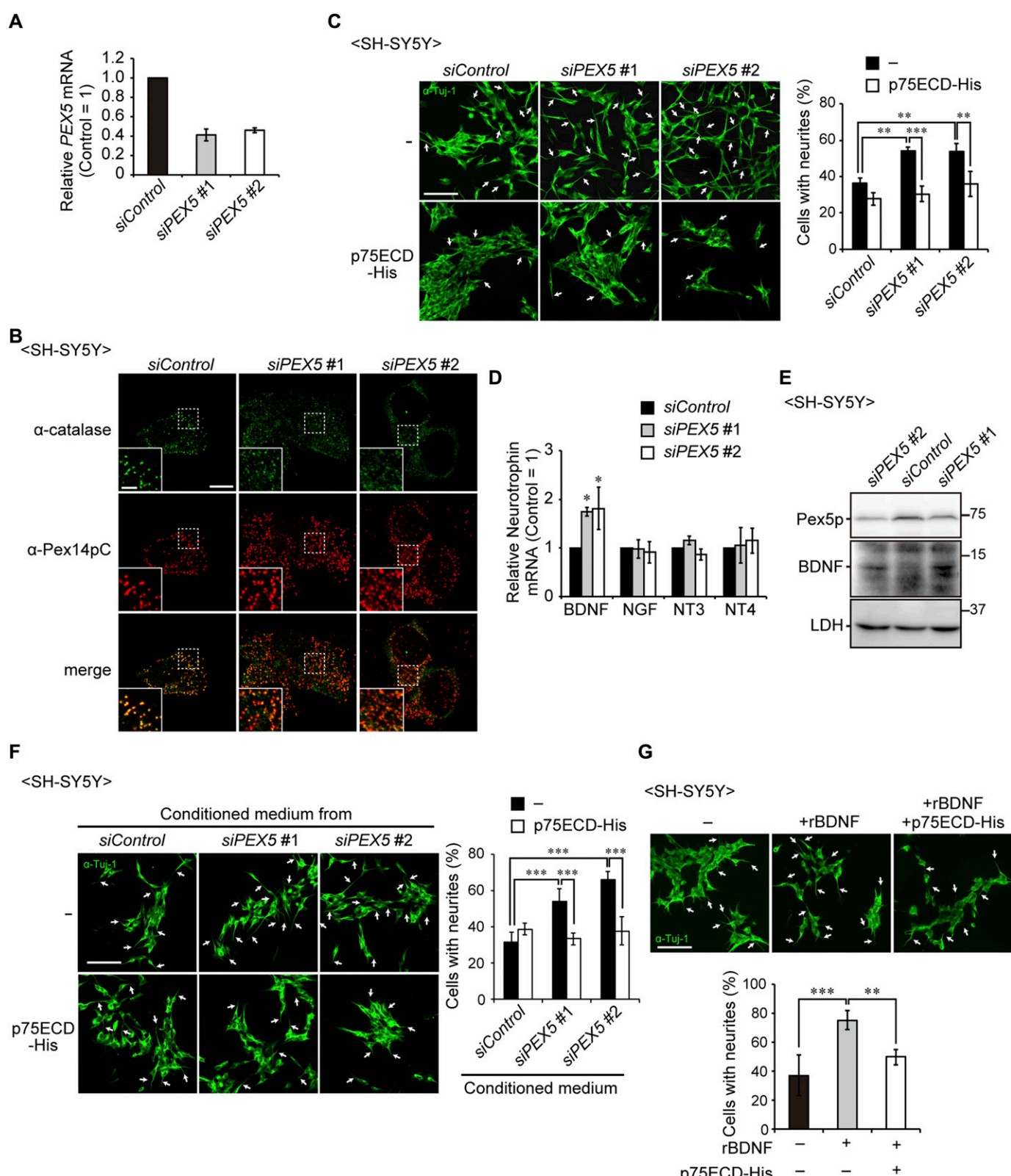

**Figure 4. Up-regulation of BDNF induces the neurite outgrowth of SH-SY5Y cells.**
**(A)** SH-SY5Y cells were treated with a control siRNA (*siControl*) or siRNAs against *PEX5* (#1 and #2) and cultured for 48 h. *PEX5* mRNA level was determined by real-time PCR (n = 3). **(B)** Cells were stained with anti-catalase (green) and Pex14pC (red) antibodies. Scale bar, 10 µm. Higher magnification images of the boxed regions are shown (inset). Scale bar, 2 µm. **(C)** SH-SY5Y cells treated with siRNAs were cultured in the presence or absence of the recombinant extracellular domain of

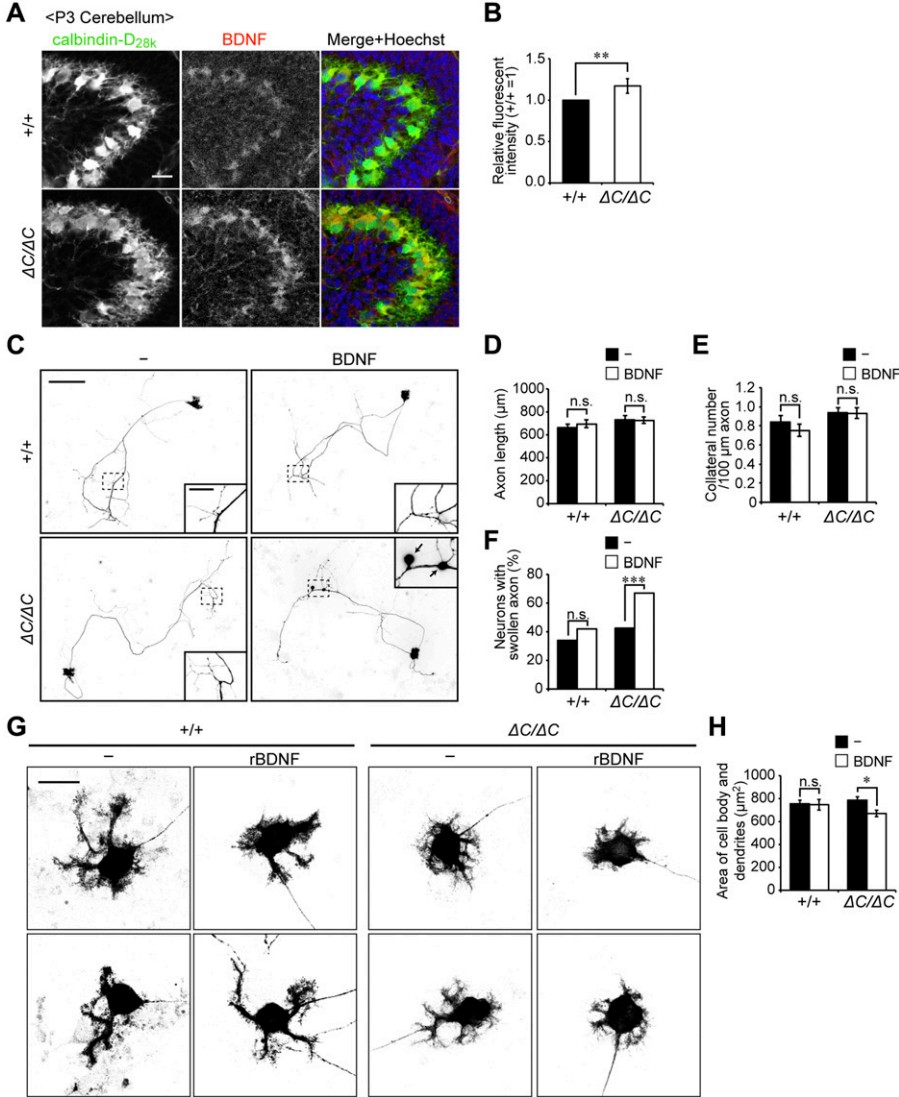

**Figure 5. Axonal swelling and impairment of dendritic development in Purkinje cells from $Pex14^{\Delta C/\Delta C}$ BL/ICR mouse upon treatment with BDNF.** **(A)** Sagittal sections of the cerebellum from wild-type (+/+) and $Pex14^{\Delta C/\Delta C}$ ($\Delta C/\Delta C$) BL/ICR mice were labeled with anti-calbindin-$D_{28k}$ (left panels, green) and BDNF (middle panels, red) antibodies. Merged views of the two different proteins and staining with Hoechst 33242 (blue) are shown on the right. Scale bar, 50 $\mu$m. **(B)** Relative fluorescent intensity of BDNF was quantified (n = 4). **(C)** Primary cerebellar neurons were cultured in the absence or presence of 50 ng/ml BDNF for 14 DIV. The cells were fixed and stained with antibody against calbindin-$D_{28K}$. Reverse images of Purkinje cells are shown. Scale bar, 100 $\mu$m. Higher magnification image of the boxed region is shown in the inset. Arrows indicate swollen axons of Purkinje cell. Scale bar, 20 $\mu$m. **(D, E)** Statistical analyses were performed for neuronal axon length (D) and the number of collaterals per 100-$\mu$m axon (E). Data represent means ± SEM. **(F)** The percentage of swollen axons (>2 $\mu$m diameter) was quantified. **(G)** Enlarged view of the reverse images of Purkinje cell bodies are shown (upper and lower panels). Scale bar, 20 $\mu$m. **(H)** Areas of the cell body and dendrites of Purkinje cells were measured (+/+: n = 64; +/+, rBDNF: n = 42; $\Delta C/\Delta C$: n = 110; $\Delta C/\Delta C$, rBDNF: n = 109). Data represent means ± SEM. ns, not significant, *$P$ < 0.05, **$P$ < 0.01, ***$P$ < 0.001, by $t$ test (B), Tukey–Kramer test (D, E, H), and $\chi^2$ test (F).

dendritic compartment of Purkinje cells. Contrary to these observations, TrkB-positive punctate structures were apparently decreased in the $Pex14^{\Delta C/\Delta C}$ cerebellum at P3 (Fig 6A, B, and D) and P7 (Fig S3A). vGlut2-positive dots were reduced in number around Purkinje cells at P3 (Fig 6C) and remained on the IGL side of cell soma at P7 in $Pex14^{\Delta C/\Delta C}$ BL/ICR mice (Fig S3B). Given that TrkB activity is thought to be involved in the promotion of CF–PC synaptic formation and stabilization (Sherrard et al, 2009), these results suggested that the deficiency of peroxisomal biogenesis gave rise

to perturbation of the CF–PC synapse through dysregulation of the BDNF-TrkB signaling pathway.

Two splicing variants of *TrkB* are abundantly expressed in the brain, and each variant functions in different cellular processes (Klein et al, 1990). Upon activation, full-length TrkB with its tyrosine kinase domain (TrkB-TK+) facing to the cytosol stimulates the MAPK/ERK, PI3K/AKT, and PLC$\gamma$ pathways that regulate neuronal survival and differentiation (Numakawa et al, 2010). By contrast, the cytosolic domain-truncated TrkB isoform (TrkB-T1) induces

p75NTR (p75ECD-His, amino acid sequence at 1–747) and stained with anti-Tuj-1 antibody (green). Arrows indicate neurons with neurite outgrowth. Percentage of neurons with neurite outgrowth were determined and shown on the right (n > 100 cells; three cultures each). Scale bar, 100 $\mu$m. **(D)** mRNA levels of neurotrophins in SH-SY5Y cells were assessed by real-time PCR (n = 3). **(E)** SH-SY5Y cell lysates were analyzed by SDS-PAGE and immunoblotting with antibodies against Pex5p, BDNF, and LDH. **(F)** Conditioned medium was obtained from the culture of SH-SY5Y cells treated with siRNAs that had been cultured for 2 d. SH-SY5Y cells were incubated in the collected conditioned medium in the presence or absence of p75ECD-His for 2 d. Percentages of neurons with neurite outgrowth were determined and shown on the right (n > 100 cells; three cultures each). Scale bar, 100 $\mu$m. **(G)** SH-SY5Y cells were cultured in the presence or absence of recombinant BDNF (rBDNF) and p75ECD-His. Cells were stained with anti-Tuj-1 antibody (green, upper panels). Percentages of neurons with neurite outgrowth were shown in a lower panel (n > 100 cells; five cultures each). Scale bar, 100 $\mu$m. Data represent means ± SD. *$P$ < 0.05, **$P$ < 0.01, ***$P$ < 0.001, by Tukey–Kramer test (C, F, G) and Dunnett's test (D).
Source data are available for this figure.

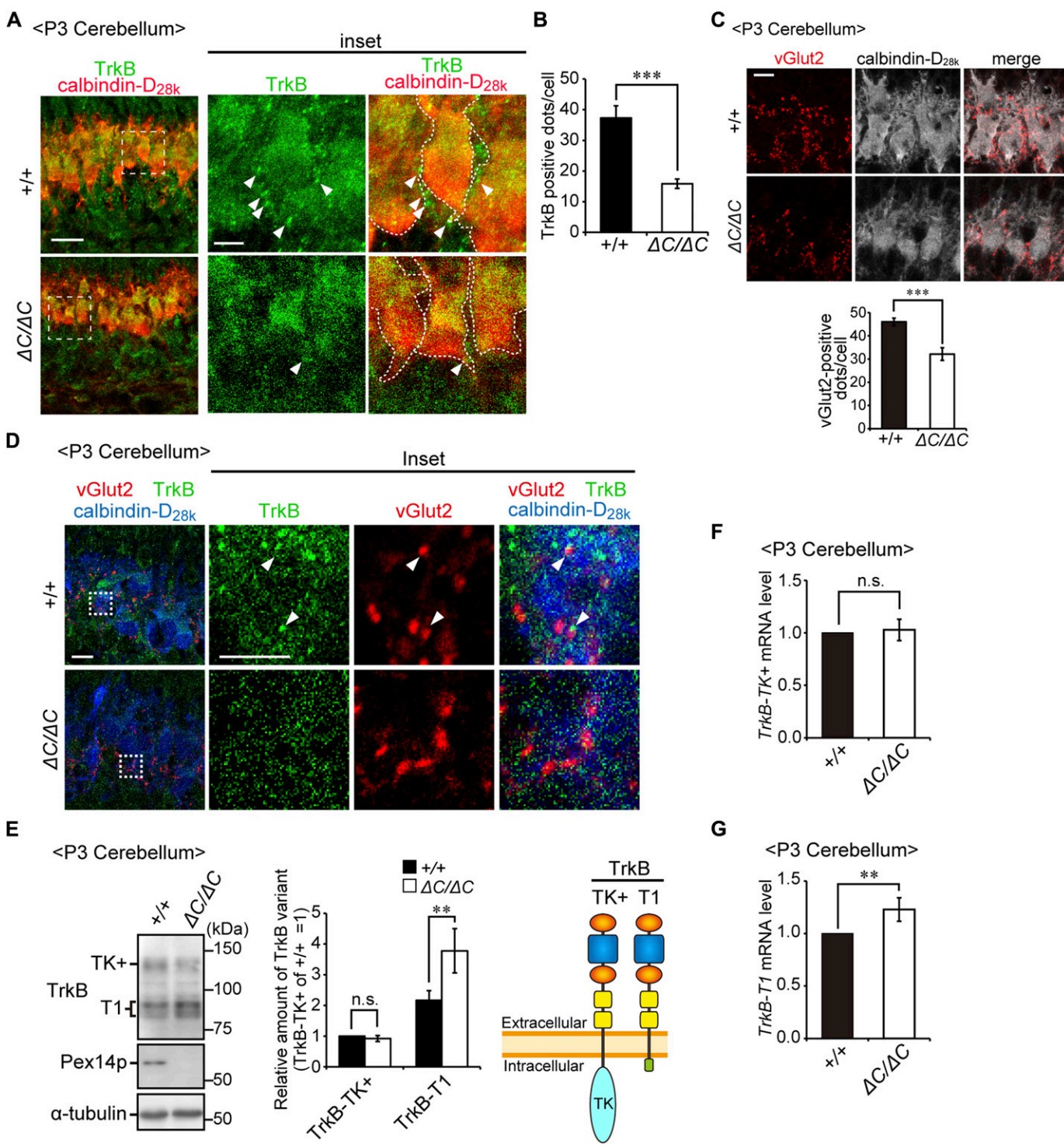

**Figure 6. TrkB-T1 is upregulated in cerebellum of *Pex14^(ΔC/ΔC)* BL/ICR mouse at P3.**
**(A)** Sagittal sections of the cerebellum from wild-type (+/+) and *Pex14^(ΔC/ΔC)* (ΔC/ΔC) BL/ICR mice were labeled with anti-TrkB (green) and calbindin-D$_{28k}$ (red) antibodies. Scale bar, 20 $\mu$m. Higher magnification images of the boxed regions are shown and cell boundaries were indicated as a dashed line (inset). Scale bar, 5 $\mu$m. **(B)** TrkB-positive dots (arrowheads) were quantified (n = 13). **(C)** Sagittal sections of the cerebellum at P3 were stained with antibodies against vGlut2 (red) and calbindin-D$_{28k}$ (white). The number of dots stained with anti-vGlut2 antibody was quantified (lower panel, n = 4). Scale bar, 10 $\mu$m. **(D)** Sagittal sections of the cerebellum at P3 were stained with antibodies against TrkB (green), vGlut2 (red), and calbindin-D$_{28k}$ (blue). Scale bar, 10 $\mu$m. Higher magnification images of the boxed regions are shown (inset). Scale bar, 5 $\mu$m. Arrowheads indicate TrkB-positive punctate structures that partly coincided with or are located adjacent to vGlut2-positive punctate structures. **(E)** Cerebellum lysates were analyzed by SDS-PAGE and immunoblotting with antibodies against TrkB, Pex14pC, and $\alpha$-tubulin (left panel). TrkB-TK+, full-length TrkB; TrkB-T1, a truncated isoform of TrkB. Amounts of TrkB-TK+ and TrkB-T1 are presented relative to those of TrkB-TK+ in the control mice (middle panel, n = 6).

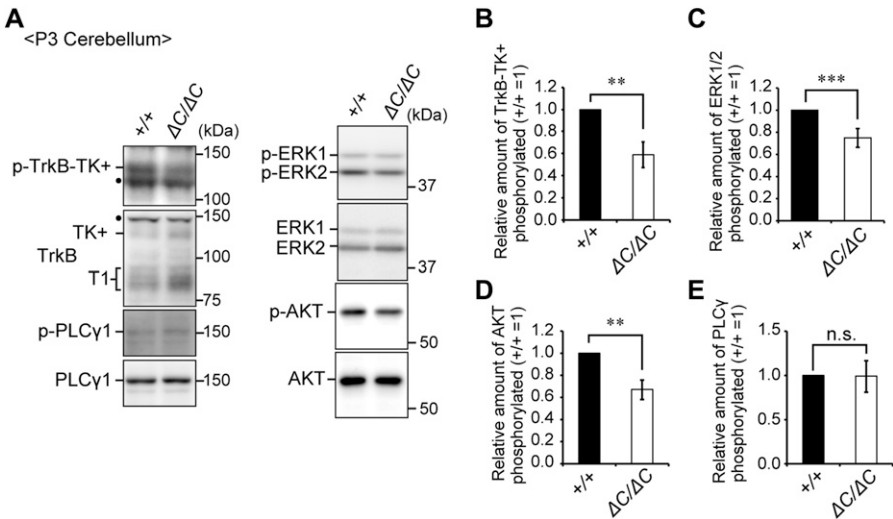

**Figure 7. The BDNF-TrkB signaling pathway is impaired in the cerebellum of Pex14^{ΔC/ΔC} BL/ICR mice at P3.**

**(A)** Cerebellum lysates from wild-type (+/+) and Pex14^{ΔC/ΔC} (ΔC/ΔC) BL/ICR mice were analyzed by SDS–PAGE and immunoblotting with antibodies against TrkB, phosphorylated Trk (p-TrkB-TK+, Y496), PLCγ1, phosphorylated PLCγ1 (p-PLCγ1, Y783), ERK, phosphorylated ERK (p-ERK1 and 2, T202 and Y204, respectively), AKT, and phosphorylated AKT (p-AKT, S473). Dots, non-specific bands. **(B–E)** The amount of phosphorylated TrkB-TK+ to total TrkB-TK+ (B), phosphorylated ERK1/2 relative to total ERK1/2 (C), phosphorylated AKT to total AKT (D), and phosphorylated PLCγ1 to total PLCγ1 (E) were represented (n = 3). ns, not significant, **P < 0.01, ***P < 0.001, by t test (B–E). Source data are available for this figure.

dominant-negative inhibition of TrkB-TK+ signaling and negatively regulates cytoskeletal rearrangement (Eide et al, 1996; Fryer et al, 1997; Fenner, 2012). We analyzed the expression level of TrkB variants in the cerebellum at P3. Immunoblotting analysis showed that TrkB-T1 was significantly elevated in the cerebellum of Pex14^{ΔC/ΔC} BL/ICR mice relative to that of wild-type mice (Fig 6E). However, there was no difference in the expression level of TrkB-TK+ between the mice of each genotype (Fig 6E and F). The up-regulation of TrkB-T1 was likely caused by increased TrkB-T1 transcription (Fig 6G). Because BDNF treatment did not influence the expression of TrkB variants in the primary culture condition, TrkB-T1 elevation is apparently in a manner independent of the elevation of BDNF in the cerebellum (Fig S3C and D). To investigate whether TrkB-TK+ signaling was attenuated in the cerebellum of Pex14^{ΔC/ΔC} BL/ICR mice, we investigated the phosphorylation level of TrkB-TK+, ERK1/2, AKT, and PLCγ. Phosphorylated TrkB-TK+, ERK1/2, and AKT were decreased in the cerebellum of the Pex14^{ΔC/ΔC} BL/ICR mouse (Fig 7A–D) and expression of c-fos and c-jun, target genes of the BDNF-TrkB signaling pathway, was reduced as well (Fig S3E), suggesting the suppression of the TrkB-TK+ signaling pathway. The elevated level of TrkB-T1 and the deactivation of TrkB-TK+ and ERK signaling pathway were evident at P5 and P7 (Fig S3F–I). On the other hand, the phosphorylation level of PLCγ was very low and not altered in Pex14^{ΔC/ΔC} BL/ICR mice (Fig 7A and E). Actin cytoskeleton was assessed by phalloidin-TRITC staining. In the wild-type mice, actin-positive structures were aligned on the dendritic tree of Purkinje cells (Fig S3J and K, upper panels). The number of actin-positive structures was reduced in Pex14^{ΔC/ΔC} BL/ICR mice (Fig S3J and K, lower panels), suggesting the impaired actin-based cytoskeletal organization in the Purkinje cells. Taken together, these results suggest that peroxisome deficiency most likely induces the up-regulation of TrkB-T1 in Purkinje cells, leading to impairment of

BDNF-TrkB-TK+ signaling in the presence of an elevated level of BDNF, and subsequently, the malformation of Purkinje cells.

## Up-regulation of BDNF expression in Pex14-deficient mice

We attempted to address how the protein level of BDNF is elevated in the cerebellum of Pex14^{ΔC/ΔC} BL/ICR mice. The Bdnf mRNA level was not altered in the cerebellum (Fig 8A). The level of mRNA for NT-4, another ligand for TrkB, was much lower in the cerebellum (Fig 8A). Immunofluorescent microscopy using antibodies to glial fibrillary acidic protein (GFAP) revealed that GFAP-positive Bergmann glia cells were not co-localized with BDNF (Fig 8B). Therefore, cerebellar glia cells are less likely to be involved in the up-regulation of BDNF in Pex14^{ΔC/ΔC} BL/ICR mice. In addition, the cells expressing both GFAP and BDNF were not detectable in the cortex at P7 (data not shown). By contrast, mRNA and protein levels of BDNF were up-regulated in ION of both genetic backgrounds of Pex14^{ΔC/ΔC} mutant mice (Fig 8C–H). Given these results, together with the fact that climbing fiber projects from ION to Purkinje cells (Sherrard & Bower, 2002), we suggest that the elevated BDNF protein in the Purkinje cell layer is most likely delivered by climbing fibers.

## Expression of TrkB variants and BDNF is not altered in the cortex of Pex14^{ΔC/ΔC} mice

ZS patients and Pex-knockout mice manifest neuronal migration defect in the cortex (Baes et al, 1997; Evrard et al, 1978; Faust & Hatten, 1997; Maxwell et al, 2003) (Fig 1F and G). TrkB was reported to be involved in the neuronal migration (Medina et al, 2004). We therefore analyzed the expression of TrkB variants in the cortex at P0.5 because cortical neuronal migration takes place from the

---

A schematic view of TrkB variants is shown on the right (right panel). TK, tyrosine kinase domain. **(F, G)** mRNA levels of TrkB-TK+ (F) and TrkB-T1 (G) were quantified by real-time PCR (n = 6). ns, not significant, **P < 0.01, ***P < 0.001, by t test (B, C, E–G). Source data are available for this figure.

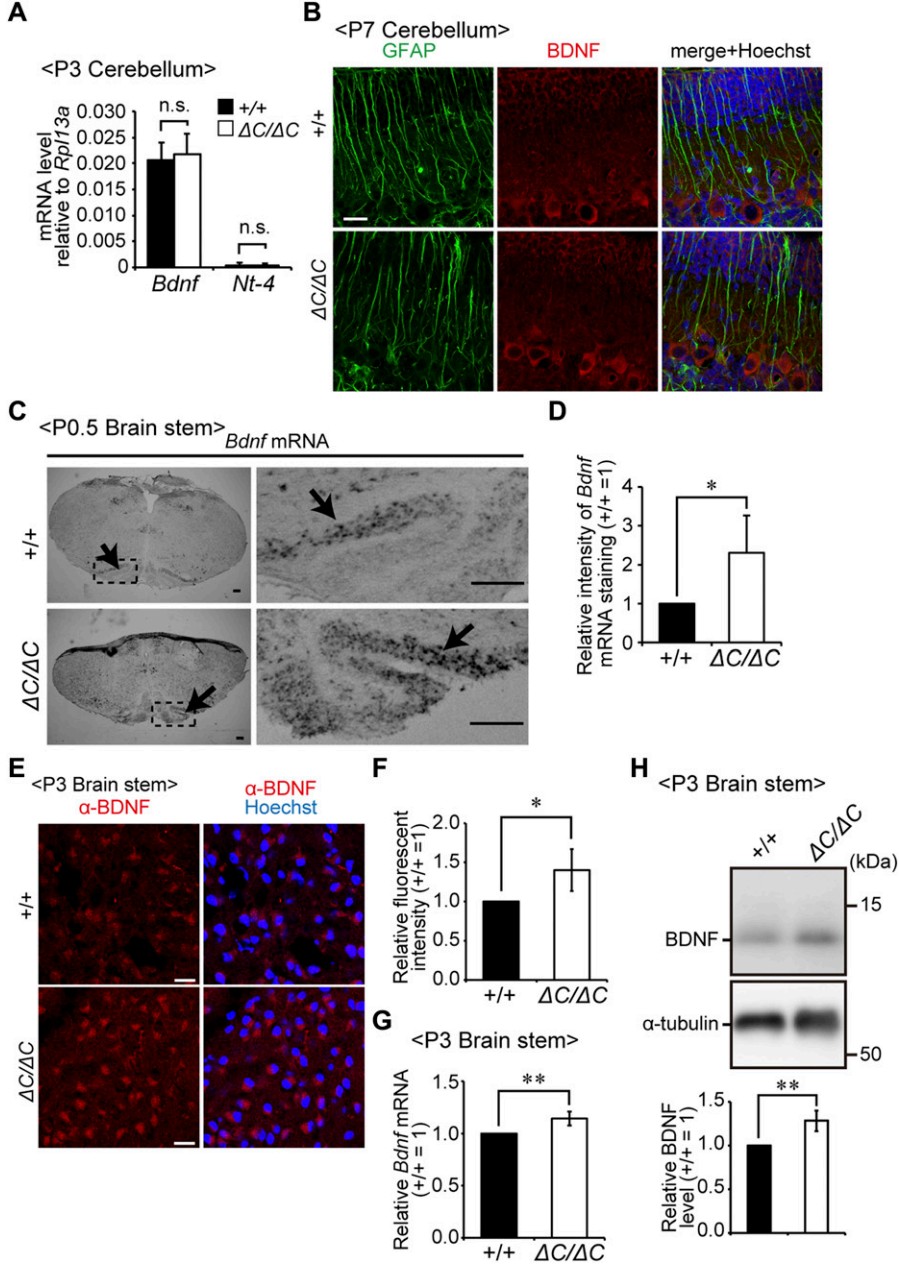

**Figure 8. BDNF expression in the cerebellum and brain stem region.**
**(A)** mRNA levels of *Bdnf* and *Nt-4* relative to those of *Rpl13a* in the cerebellum from wild-type (+/+) and *Pex14*[ΔC/ΔC] (ΔC/ΔC) BL/ICR mice at P3 were determined by real-time PCR (n = 4). **(B)** Sagittal sections of the cerebellum at P7 were stained with antibodies to GFAP (green) and BDNF (red). Merged views of the two different proteins and staining with Hoechst 33242 (blue) are shown. Scale bar, 20 μm. **(C)** In situ hybridization analysis of *Bdnf* mRNA in the brain stem region of wild-type (upper panels) and *Pex14*[ΔC/ΔC] (lower panels) mice at P0.5. Arrows indicate ION. Insets are higher magnification images of the dashed-line boxed regions. Scale bar, 100 μm. **(D)** Relative intensity of *Bdnf* mRNA staining by in situ hybridization shown in C was presented (n = 3). **(E)** Sagittal sections of the brain stem at P3 were stained with anti-BDNF antibody (red) and Hoechst 33242 (blue). Scale bar, 20 μm. **(F)** Fluorescent intensity of BDNF staining on the ION shown in E was quantified (n = 3). **(G)** The mRNA level of *Bdnf* in the brain stem region was quantified by real-time PCR (n = 3). **(H)** Brain stem lysates were analyzed by SDS–PAGE and immunoblotting using antibodies against BDNF and α-tubulin. The BDNF band was quantified (lower panel, n = 4). ns, not significant, *P < 0.05, **P < 0.01, by t test (A, D, F–H). Source data are available for this figure.

embryo to early postnatal period. In the cortex, TrkB-TK+ was abundantly expressed and there was no difference in the expression of TrkB variants and BDNF between wild-type and *Pex14*[ΔC/ΔC] mice (Fig S4A–C). In addition, phosphorylation levels of ERK and AKT were not altered (Fig S4D and E). These results suggested that the BDNF-TrkB signaling pathway is not responsible for the neuronal migration defect in the cortex of *Pex14*[ΔC/ΔC] mice.

## Discussion

The pathological mechanisms underlying the malformation of the CNS in patients with ZSDs are undefined. In the present study, we

established a new mouse model of ZS, *Pex14*-defective mouse showing mild defect of peroxisomal protein import and metabolisms. At an early stage of cerebellum development in *Pex14*[ΔC/ΔC] BL/ICR mice, the increase in BDNF together with an aberrant expression of TrkB-T1 and dysregulation of BDNF-TrkB signaling were induced. Such findings suggest that impairment in the BDNF-TrkB signaling pathway is involved in the dysmorphogenesis of the cerebellum in ZSDs.

In *Pex14*[ΔC/ΔC] mice, import of PTS1 and PTS2 proteins is partially impaired, whereas catalase import is completely defective (Fig 2A–E). The targeting signal of catalase is noncanonical PTS1 sequence, KANL, instead of canonical SKL motif, at the C-terminus (Otera & Fujiki, 2012; Purdue & Lazarow, 1996). Binding affinity of the

targeting signal of catalase to the cytosolic receptor, Pex5p, is weaker than the canonical PTS1 proteins (Otera & Fujiki, 2012). In addition, catalase is imported into peroxisomes as a tetrameric complex formed in the cytosol and its oligomeric import is dependent on the Pex5p-Pex13p interaction (Otera & Fujiki, 2012). Catalase import is more susceptible to the impairment of peroxisomal matrix import than canonical PTS1 proteins, as observed in the fibroblasts from the patients with IRD, less severe ZSD (Tamura et al, 2001). In *Pex14$^{\Delta C/\Delta C}$* mice, Pex14p is truncated in the C-terminal half part (Fig 1C). Because the Pex5p-binding region comprising amino-acid sequence at 25 to 70 remained in Pex14p of *Pex14$^{\Delta C/\Delta C}$* mouse, Pex5p–cargo complex could target the surfaces of peroxisomes, giving rise to partial import of matrix proteins. Moreover, the coiled-coil domain of Pex14p is deleted in *Pex14$^{\Delta C/\Delta C}$* mice. We earlier reported that the coiled-coil domain of Pex14p is required for homo-oligomerization of Pex14p (Itoh & Fujiki, 2006). In the fibroblasts from the patient with Pex14p-Q185X mutation (Shimozawa et al, 2004), where the coiled-coil domain is truncated, catalase import is severely impaired but PTS1 import is partially affected (Fig S1). These results suggest that the coiled-coil region-dependent homo-oligomerization of Pex14p is required for the efficient import of peroxisomal matrix proteins, particularly the oligomeric import of catalase.

The *Pex14$^{\Delta C/\Delta C}$* mouse shows a typical ZS phenotype, such as growth retardation, death shortly after birth, neuronal migration defects, and malformation of the cerebellum. However, the impaired peroxisome matrix protein import and peroxisomal metabolism (Fig 2) are milder than those observed in other *Pex*-knockout mice (Baes et al, 1997; Faust & Hatten, 1997; Janssen et al, 2003; Maxwell et al, 2003). The metabolic abnormalities and defects in protein import in fibroblasts from a patient with a *Pex14* mutation (Pex14p-Q185X) (Shimozawa et al, 2004) are similarly milder than those in fibroblasts from ZS patients (Abe et al, 2014) (Fig S1). Therefore, a mild defect in peroxisomal metabolism could also cause the severe CNS defects observed in the most severe ZS cases.

In the early stage of cerebellar development, multiple climbing fibers from the ION attach to the cell bodies of Purkinje cells. This is followed by the translocation of one climbing fiber to the dendritic tree and the establishment of climbing fiber input together with the pruning of the other climbing fibers (Sugihara, 2005; Watanabe & Kano, 2011). Spatiotemporal BDNF-TrkB signaling is essential for the development of the cerebellum, especially in CF–PC synapse formation at an early developmental stage (Sherrard & Bower, 2002), pruning of the CF–PC synapse (Johnson et al, 2007), and Purkinje cell arborization, as demonstrated in both types of *Bdnf*- (Carter et al, 2002; Schwartz et al, 1997) and *TrkB-TK+*- (Minichiello & Klein, 1996; Rico et al, 2002) knockout mice. Apparent reduction of CF–PC synapse and the defect of pruning were observed in *Pex14$^{\Delta C/\Delta C}$* BL/ICR mice (Figs 6C and S3B). The reduction of the CF–PC synapse in peroxisome-deficient mice was also reported in the *Pex2$^{-/-}$* mouse cerebellum, whereas the positioning of climbing fiber to Purkinje cell layer was normal (Faust, 2003). These results imply the impaired maturation of CF–PC synapse. It is conceivable that TrkB-TK+ is located on the post-synaptic compartment of the dendritic spine and is involved in synapse formation and stabilization (Yoshii & Constantine-Paton, 2010). In addition, BDNF-TrkB activity is thought to be involved in the promotion and stabilization of CF–PC

synaptic formation (Sherrard et al, 2009). Thus, severe reduction of TrkB-positive punctate structures in *Pex14$^{\Delta C/\Delta C}$* BL/ICR mice (Fig 6A and B) could be explained by the excess TrkB-T1 that affects the expression of TrkB-TK+ at the cell surface (Haapasalo et al, 2002), thereby suppressing the phosphorylation of TrkB-TK+ and its downstream signaling targets, ERK and AKT (Fig 7). Based on these findings, it is most likely that the disturbance of the BDNF-TrkB signaling pathway in the cerebellum is involved in the impairment in CF–PC synapse maturation.

In the developmental stage of the cerebellum, BDNF immunoreactivity is evident around Purkinje cells (Schwartz et al, 1997), whereas expression of *Bdnf* mRNA is undetectable using either Northern blot analysis (Maisonpierre et al, 1990) or in situ hybridization (Rocamora et al, 1993). Therefore, the BDNF protein in the Purkinje cell layer is thought to be delivered via climbing fibers from the ION (Lindholm et al, 1997; Sherrard & Bower, 2002), where *Bdnf* mRNA is strongly expressed at the stage of climbing fiber growth and synaptogenesis (Rocamora et al, 1993). Indeed, the expression of *Bdnf* mRNA is up-regulated in the ION neurons of *Pex14$^{\Delta C/\Delta C}$* mice (Fig 8C–H). However, it remains undefined how the elevation of *Bdnf* mRNA is regulated in the peroxisome-deficient ION neurons. Further studies would be required to elucidate the molecular mechanisms underlying the transcriptional up-regulation of the *Bdnf* gene in the ION of *Pex*-knockout mice.

In *Pex14$^{\Delta C/\Delta C}$* BL/ICR mouse cerebellum, Purkinje cells show axonal swelling and growth defects of dendrites. These defects of neuronal morphogenesis in the cerebellum of *Pex14$^{\Delta C/\Delta C}$* mice are likely explained by the impaired spatiotemporal BDNF-TrkB signaling pathway apparently induced by excess BDNF, not NT-4 (Fig 8A), and enhanced expression of TrkB-T1 (Fig 6E and G). Indeed, in the primary culture of cerebellar neurons, the dendritic development of Purkinje cells from *Pex14$^{\Delta C/\Delta C}$* BL/ICR mice was compromised (Fig 5G and H) and swollen axons were increased by the BDNF treatment (Fig 5C and F). The defect of dendritic arborization in primary Purkinje cells is in good agreement with the notion that excess BDNF inhibits the dendritic growth of the cells expressing TrkB-T1 via the dominant-negative inhibition of TrkB-TK+ signaling (Fenner, 2012; Yacoubian & Lo, 2000), although the downstream of the BDNF signaling pathway of dendritic abnormality and its relationship to the signaling for axonal swelling remain unclear. There is no effect of BDNF on the wild-type Purkinje cells in the primary culture of cerebellar cells (Fig 5), consistent with the finding that BDNF-transgenic mice show no significant difference of cerebellar morphology (Bao et al, 1999). Therefore, the malformation of Purkinje cells in *Pex14$^{\Delta C/\Delta C}$* BL/ICR mice is most likely caused by a combination of elevated BDNF and prominent expression of TrkB-T1 (Fig S5). The findings in the in vitro assay of primary cerebellar neurons could also exclude the possibility that only growth retardation caused by the deficiency of peroxisomal functions in liver compromises the maturation of the cerebellum (Krysko et al, 2007).

The molecular mechanism underlying the alternative splicing of the *TrkB-T1* variant remains elusive. In the *Pex14$^{\Delta C/\Delta C}$* BL/ICR mouse cerebellum, TrkB-T1 expression is up-regulated and phosphorylation of TrkB-TK+, ERK, and AKT is down-regulated, suggesting the inhibition of TrkB-TK+-elicited signal transduction. The up-regulation of TrkB-T1 is less likely dependent on the elevation of

BDNF in $Pex14^{\Delta C/\Delta C}$ BL/ICR mice because BDNF treatment did not alter the TrkB-T1 expression in the primary neuron (Fig S3C and D). In addition, the BDNF transgenic mouse shows normal cerebellar development (Bao et al, 1999), suggesting that excess BDNF is irrelevant to up-regulation of TrkB-T1 (Bao et al, 1999). The elevation of mRNA for *TrkB-T1* in the $Pex14^{\Delta C/\Delta C}$ BL/ICR mouse cerebellum suggests that *TrkB* mRNA splicing is affected by the defect in peroxisomal biogenesis. Therefore, molecular mechanisms addressing the issue of how peroxisomal dysfunction affects the splicing of *TrkB* in Purkinje cells will shed light on understanding the malformation of the cerebellum in PBDs.

In the cerebellum of patients with ZS, dysmorphology of the Purkinje cell arborization, heterotopia of Purkinje cells in the white matter, and granule cell clustering between the Purkinje cells are observed (Volpe & Adams, 1972; de León et al, 1977; Evrard et al, 1978; Powers & Moser, 1998; Crane, 2014). These phenotypes resemble those in *Pex*-knockout mice, including the impairment of Purkinje cell arborization and granule cell migration defect, although the heterotopic Purkinje cells are much less severe (Faust & Hatten, 1997; Faust, 2003). During cerebellar development, the BDNF-TrkB signaling pathway plays pivotal roles in Purkinje cell arborization (Minichiello & Klein, 1996; Schwartz et al, 1997; Carter et al, 2002; Rico et al, 2002) and in granule cell migration from the external granular layer to the IGL (Zhou et al, 2007). Heterotopic Purkinje cells are also observed in the patients with milder ZSDs, NALD, and IRD (Aubourg et al, 1986; Torvik et al, 1988; Chow et al, 1992). Therefore, the BDNF-TrkB signaling pathway in the cerebellum is most likely susceptible to the impaired peroxisomal metabolism in ZSDs, including ZS, NALD, and IRD.

# Materials and Methods

## Construction of a targeting vector and generation of $Pex14^{\Delta C/\Delta C}$ mice

Genomic DNA corresponding to the *Pex14* locus was isolated from 129/Sv mouse genomic library (Agilent Technologies). The targeting vector was constructed by replacing a 6.5-kb SacI-AorH51I fragment of genomic DNA containing three *Pex14* exons, exons 6–8, with a PGK-lox-neo-poly (A) cassette. The vector thus contained 1.7- and 6.0-kb regions of homology located 5′ and 3′, respectively, relative to the neomycin resistance gene (Neo$^r$). A MC1-DT-A-poly (A) cassette was ligated at the 3′ end of the targeting construct. The maintenance, transfection, and selection of ES cells derived from 129/Sv mice were carried out as described previously (Nakayama et al, 1996). Mutant ES cells were microinjected into C57BL/6 mouse blastocysts, and the resulting male chimeras were mated with C57BL/6 females. Germ line transmission of the mutant allele was confirmed by Southern blot analysis. Heterozygous offspring were intercrossed to produce homozygous mutant animals. PCR analysis of tail biopsy genomic DNA was undertaken using primers P14F (5′-GTATAAATGTGGGAGTTTCCCTGG-3′) and P14R (5′-GTACTTGTGAACTC-TGCTGGTAC-3′) to amplify a 599-bp fragment specific for the wild-type allele and primers P14F and KN52-2 (5′-GTGTTGGGTCG-TTTGTTCGG-3′) to amplify a 169-bp fragment specific for the disrupted *Pex14* gene.

## Antibodies and plasmids

Mouse monoclonal antibodies against calbindin-D$_{28k}$ (CB-955) and GFAP (G-A-5) were purchased from Sigma-Aldrich, and mouse monoclonal antibody to α-tubulin was from BD Biosciences. Rabbit antibodies to BDNF (N-20) and TrkB (H-181) and mouse antibody to phosphorylated Trk (Y496, E-6) were from Santa Cruz Biotechnology. Mouse monoclonal antibodies against Tuj-1 and ERK1/2 and rabbit antibody to phospho-ERK1/2 were from R&D systems. Rabbit antibodies to AKT (N3C2), phosphor-AKT (S473), PLCγ, and phosphor-PLCγ (Y783) were from GeneTex. Guinea pig antibody to vGlut2 and goat antibody to lactate dehydrogenase (LDH) were from Millipore and Rockland, respectively. We used rabbit antisera against PTS1 peptides (Otera et al, 1998), rat AOx (Tsukamoto et al, 1990), human catalase (Shimozawa et al, 1992), rat catalase (Tsukamoto et al, 1990), mouse ADAPS (Honsho et al, 2008), rat 70-kD peroxisomal integral membrane protein (PMP70) (Tsukamoto et al, 1990), Chinese hamster Pex5p (Otera et al, 2000), and the carboxyl terminal part of rat Pex14p (Pex14pC) (Shimizu et al, 1999). Rabbit antibody raised to amino-terminal 106 amino acid residues of rat Pex14p (Pex14pN) (Okumoto et al, 2000; Itoh & Fujiki, 2006) was also used. The p75ECD plasmid coding for the amino-acid sequence at 1–747 was ligated into the HindIII-XhoI sites of pSecTag2/Hygro C vector (Invitrogen), yielding pSecTag2/*p75ECD-His*.

## Cell culture

CHO K1 cells were cultured in Ham's F12 medium (Invitrogen) containing 10% FBS. SH-SY5Y from the Human Science Research Resources Bank and fibroblasts from a control and a patient with *PEX14* mutation (K01) (Shimozawa et al, 2004) were cultured in DMEM (Invitrogen) supplemented with 10% FBS. Transfection of pSecTag2/*p75ECD-His* to CHO-K1 was carried out with Lipofect-amine (Invitrogen) according to the manufacturer's instructions. Stable transformants of CHO K1 cells were isolated by selection with 1.2 μg/ml hygromycin (Sigma-Aldrich). For siRNA transfection, SH-SY5Y cells ($2 \times 10^5$) were transfected with 5 pmol dsRNA using Lipofectamine RNAiMAX (Invitrogen) and plated on poly-L-lysine (Sigma-Aldrich)–coated cover glasses. The sequences siRNA used were as follows: *PEX5* #1, 5′-UAAUAGUUCUUGUUCAUUCUCUGCC-3′ and 5′-GGCAGAGAAUGAACAAGAACUAUUA-3′; *PEX5* #2, 5′-UUUAG-CUCCAGACACCUCCGCAAUG-3′ and 5′-CAUUGCGGAGGUGUCUGGAGC-UAAA-3′. Cells were cultured for 2 d and stained with antibodies to Tuj-1 and Hoechst 33242.

## Primary culture of cerebellar neurons

Primary cerebellar neurons were prepared from a mouse at P0.5. Briefly, cerebella were excised into small pieces and dissociated with 15 units of papain (Worthington Biochemical Corporation) in dissociation solutions (0.2 mg/ml L-cysteine, 0.2 mg/ml BSA, and 10 mg/ml glucose) and 0.1 mg/ml DNase I (Sigma-Aldrich). Cells were separated by gentle trituration passes using a 10-ml pipette and were passed through a 70-μm cell strainer (BD Biosciences) to remove large debris. Dissociated cerebellar cells ($3.0 \times 10^5$ cells/cm$^2$) were plated at 37°C under 5% CO$_2$ on poly-L-lysine– and laminin (Sigma-Aldrich)–coated cover glasses in Neurobasal

medium (Invitrogen) containing B27 supplement (Invitrogen) and 0.5 mM L-glutamine. Culture medium was replaced with fresh medium every 3–4 d. Purkinje cells were stained with anti-calbindin-D$_{28k}$ antibody and observed by an AF 6000LX microscope (Leica). The areas of cell body and dendrites of Purkinje cells were measured by Image J software (National Institutes of Health).

## Purification of p75ECD-His

A stable transformant of CHO K1 cell expressing p75ECD-His was cultured in serum-free F12 medium for 3 d. The cell cultured medium was collected and centrifuged to remove floating cells. The resulting supernatant fraction was mixed and incubated with Ni-NTA agarose beads (QIAGEN) for 4 h. The p75ECD-His–bound beads were washed 6 times with purification buffer (50 mM Hepes-KOH, pH 7.4, 150 mM NaCl, 20 mM imidazole, and 10% glycerol) followed by elution with purification buffer containing 250 mM imidazole. The eluent was loaded onto a PD10 column (GE Healthcare) with suspension buffer (50 mM Hepes-KOH, pH 7.4, 150 mM NaCl, and 10% glycerol) and then concentrated by ultrafiltration in an Amicon Ultra-15 (10,000 molecular weight cut-off; Millipore). Purified p75ECD-His was added to the cell culture at 1.0 μg/ml.

## Lipid extraction

Cells were detached from culture plates by incubation with trypsin and suspended in PBS. Protein concentration was determined by the bicinchonic acid method (Thermo Fisher Scientific). Total lipids were extracted from 50 μg of total cellular proteins by the Bligh and Dyer method (Bligh & Dyer, 1959). Cells were suspended in methanol/chloroform/water at 2:1:0.8 (vol/vol/vol) and then 50 pmol of 1-heptadecanoyl-sn-glycero-3-phosphocholine (LPC; Avanti Polar Lipids), 1, 2-didodeca-noyl-sn-glycero-3-phosphocholine (DDPC; Avanti Polar Lipids), and 1, 2-didodecanoyl-sn-glycero-3-phosphoethanolamine (DDPE; Avanti Polar Lipids) were added as internal standards. After incubation for 5 min at room temperature, 1 ml each of water and chloroform was added and the samples were then centrifuged at 720 g for 5 min in Himac CF-16RX (Hitachi Koki) to collect the lower organic phase. To re-extract lipids from the water phase, 1 ml chloroform was added. The combined organic phase was evaporated under a nitrogen stream and the extracted lipids were dissolved in methanol.

## Liquid chromatography coupled with tandem mass spectrometry (LC-MS/MS)

LC-MS/MS was performed as described (Abe et al, 2014) using a 4000 Q-TRAP quadrupole linear ion trap hybrid mass spectrometer (AB Sciex) with an ACQUITY UPLC system (Waters). The data were analyzed and quantified using Analyst software (AB Sciex).

## Real-time RT-PCR

Total RNA was extracted from cells using TRIzol reagent (Invitrogen) and first-strand cDNA was synthesized by the PrimeScript RT reagent Kit (Takara Bio). Quantitative real-time RT-PCR was performed with SYBR Premix Ex Taq II (Takara Bio) using an Mx3000P QPCR

system (Agilent Technologies). Several sets of primers used are listed in Table S1.

## Immunofluorescent microscopy

Cultured cells were fixed with 4% paraformaldehyde in PBS, pH 7.4, for 15 min at room temperature. Peroxisomes were visualized by indirect immunofluorescence staining with indicated antibodies as described previously (Mukai et al, 2002). Antigen-antibody complexes were detected with goat anti-mouse and anti-rabbit IgG conjugated to Alexa Fluor 488 and Alexa Fluor 568 (Invitrogen). Phalloidin-TRITC (Sigma-Aldrich) was used for the staining of F-actin. Images were obtained using a laser-scanning confocal microscope (LSM 710 with Axio Observer.Z1; Carl Zeiss).

## Immunohistochemistry

Neonatal mice were deeply anesthetized with isoflurane and perfused transcardially with 1 ml of 4% PFA in 0.1 M phosphate buffer (PB). After decapitation, the brains were fixed in the same buffer overnight at 4°C and were transferred to 30% sucrose in PBS for 2 d. The brains were embedded in the Tissue-Tek OCT compound (Sakura Finetek) and subsequently frozen at −80°C. Cryosections were cut at a thickness of 20 μm using a cryostat Microm HM550-OMP (Thermo Fisher Scientific) and were then mounted on MAS-coated glass slides (Matsunami Glass). The sections were permeabilized by ice-cold methanol, blocked by blocking buffer (10% BSA, 0.3% Triton X-100 in PBS), and then incubated overnight at 4°C with primary antibody diluted in blocking buffer or Can Get Signal immunostain solution A (Toyobo). After washing with PBS, the sections were incubated with appropriate secondary antibody conjugated to Alexa 488 or 567; for marker staining of nuclei, the sections were incubated with Hoechst 33242 in PBS for 2 min at room temperature, and then mounted with PermaFluor. Images were obtained under an LSM 710, AF 6000LX, or BZ-9000 (Keyence) microscope. Quantitative analysis was performed by BZII-analyzer (Keyence).

## Immunoblotting

Immunoblotting was performed as previously described (Honsho et al, 2015). Precision Plus Protein All Blue standards (Bio-Rad) were used as molecular size markers. Immunoblots were developed with ECL prime reagent (GE Healthcare) and immunoreactive bands were detected by X-ray film (GE Healthcare) or LAS-4000 Mini luminescent image analyzer (Fujifilm). The band intensities were quantified by Image J software or Image Gauge software (Fujifilm).

## In situ hybridization

Brains were removed, fixed overnight in 4% PFA in 0.1 M PB, incubated another overnight in 30% sucrose/4% PFA in 0.1 M PB, and sectioned at 20 μm by a cryostat, as previously described (Ueno et al, 2012). Digoxigenin-labeled riboprobes were prepared by in vitro transcription. Sections were hybridized with digoxigenin-labeled probe at 70°C overnight. Excess probes were washed out and signals were detected with alkaline phosphatase–coupled

antibody to digoxigenin (Roche Diagnostics) with nitroblue tetrazolium and 5-bromo-4-chloro-3-indolyl phosphate as color reaction substrates.

## Statistical analysis

Statistical analysis was performed using R software (http://www.r-project.org). All *t* tests used were one-tailed. A *P* value < 0.05 was considered statistically significant. Data are shown as means ± SD unless otherwise described.

## Study approval

The animal ethics committee of Kyushu University approved all animal experiments.

# Supplementary Information

# Acknowledgments

We thank R.J.A. Wanders for providing fibroblasts from a patient with *PEX14* mutation. We also thank Y Nanri and S Okuno for technical assistance, K Shimizu for preparing the figures, and the other members of our laboratory for helpful discussion. We appreciate the technical assistance from The Research Support Center, Research Center for Human Disease Modeling, Kyushu University Graduate School of Medical Sciences and Laboratory for Technical Support, Medical Institute of Bioregulation, Kyushu University. This work was supported in part by grants from the Ministry of Education, Culture, Sports, Science, and Technology of Japan; Grants-in-Aid for Scientific Research (no. JP17K15621 to Y Abe; nos. JP24247038, JP25112518, JP25116717, JP26116007, JP15K14511, JP15K21743, and JP17H03675 to Y Fujiki); grants from the Takeda Science Foundation (to Y Fujiki), the Naito Foundation (to Y Fujiki), the Japan Foundation for Applied Enzymology, and the Novartis Foundation (Japan) for the Promotion of Science (to Y Fujiki).

## Author Contributions

Y Abe: conceptualization, resources, data curation, formal analysis, funding acquisition, validation, investigation, visualization, methodology, project administration, and writing—original draft.
M Honsho: conceptualization, resources, data curation, investigation, visualization, project administration, and writing—original draft.
R Itoh: resources and formal analysis.
R Kawaguchi: formal analysis.
M Fujitani: resources, formal analysis, validation, and investigation.
K Fujiwara: formal analysis.
M Hirokane: formal analysis.
T Matsuzaki: formal analysis and methodology.
K Nakayama: resources, formal analysis, and methodology.
R Ohgi: formal analysis.
T Marutani: resources and formal analysis.
KI Nakayama: resources, formal analysis, and methodology.
T Yamashita: resources, formal analysis, and methodology.
Y Fujiki: conceptualization, resources, data curation, supervision, funding acquisition, investigation, project administration, and writing—original draft, review, and editing.

## Conflict of Interest Statement

The authors declare no competing financial interests.

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
