## [Reviewer comments · Life Science Alliance]

Life Science Alliance

Peroxisome biogenesis deficiency attenuates the BDNF-TrkB pathway-mediated development of cerebellum

Yuichi Abe, Masanori Honsho, Ryoko Kawaguchi, Ryota Itoh, Masashi Fujitani, Kazuhiro Fujiwara, Masaaki Hirokane, Takashi Matsuzaki, Keiko Nakayama, Ryohei Ohgi, Toshihiro Marutani, Keiichi Nakayama, Toshihide Yamashita, and Yukio Fujiki

DOI: [10.26508/lsa.201800062](https://doi.org/10.26508/lsa.201800062)

Corresponding author(s): Yukio Fujiki, Kyushu University

Review Timeline:

Submission Date:	2018-04-02
Editorial Decision:	2018-06-04
Revision Received:	2018-10-22
Editorial Decision:	2018-11-05
Revision Received:	2018-11-08
Accepted:	2018-11-08

Scientific Editor: Andrea Leibfried

Transaction Report:

June 4, 2018

Re: Life Science Alliance manuscript #LSA-2018-00062-T

Prof. Yukio Fujiki
Kyushu University
International Institute for Carbon-Neutral Energy Research
744 Motooka, Nishi-ku
Higashi-ku
Fukuoka 819-0395
Japan

Dear Dr. Fujiki,

Thank you for submitting your manuscript entitled "Peroxisome biogenesis deficiency attenuates the BDNF-TrkB pathway-mediated development of cerebellum" to Life Science Alliance. Please excuse the delay in getting back to you. This delay is due to the fact that I had to seek additional input on your work during late stages of the peer review process, because a critical reviewer was unresponsive and did not provide a report. I have now three reports on your work, which are appended to this letter.

As you will see, the reviewers think that the data shown are generally well-performed. However, the reviewers also think that the *in vitro* and *in vivo* assays cannot be compared to each other and do not provide the kind of coherent, valuable study we should publish in Life Science Alliance. The fact that the Pex-14 deficient mice do not present a hippocampal phenotype nor BDNF increase in this brain area reduces the value of the *in vitro* assays using hippocampal neurons.

You will see that the reviewers provide constructive input on how to address this issue, but a lot of effort and work needs to be done to do so. Reviewer #4 proposes to rather shift the emphasis, concentrating on the cerebellum phenotype only (removal of the hippocampal part altogether), and reviewer #3 agreed during reviewer cross-commenting that this would be a good way to address the discordance of the *in vitro* and *in vivo* results.

We agree with this view and can therefore offer to invite submission of a revised version of your work. Importantly, the revised version should be clearly re-focussed on the *in vivo* part, and the abstract would need to reflect this shift. More *in vivo* evidence is required (see also reviewer #3), and the description of the phenotype and the origin of the excess BDNF potentially responsible for the phenotype needs to be improved.

- A letter addressing the reviewers' comments point by point.
- An editable version of the final text (.DOC or .DOCX) is needed for copyediting (no PDFs).
- High-resolution figure, supplementary figure and video files uploaded as individual files: See our detailed guidelines for preparing your production-ready images, <http://life-science-alliance.org/authorguide>
- Summary blurb (enter in submission system): A short text summarizing in a single sentence the study (max. 200 characters including spaces). This text is used in conjunction with the titles of papers, hence should be informative and complementary to the title and running title. It should describe the context and significance of the findings for a general readership; it should be written in the present tense and refer to the work in the third person. Author names should not be mentioned.

B. MANUSCRIPT ORGANIZATION AND FORMATTING:

Full guidelines are available on our Instructions for Authors page, <http://life-science-alliance.org/authorguide>

Thank you for this interesting contribution to Life Science Alliance. We are looking forward to receiving your revised manuscript.

Sincerely,

Andrea Leibfried, PhD
Executive Editor

Life Science Alliance
Meyerhofstr. 1
69117 Heidelberg, Germany
t +49 6221 8891 502
e a.leibfried@life-science-alliance.org
www.life-science-alliance.org

Reviewer #1 (Comments to the Authors (Required)):

In the manuscript Abe et al. study the cause for cerebellar malfunction in patients afflicted by peroxisomal biogenesis disorders (PBD). It has been known that neuronal migration is aberrant in patients but the root cause has not been determined.

The authors develop a new transgenic mouse model for PBD in which dominant negative (DN) alleles of PEX5 and PEX14 (deletions of essential domains) are expressed. With these, they first provide evidence that expression of these DN constructs in an astrocyte cell line co-cultured with hippocampal neurons cause an increase in axon collateral development. Looking for a cause, they discover that expression of the neurotropic factor Bdnf is upregulated in the glial cells. Conditioned medium from the astrocytes could reproduce the effect on collaterals, as could diluted Bdnf itself. How does a defect in peroxisomal biogenesis cause an increase in Bdnf secretion? The authors show that secretion can be reduced by the catalase inhibitor 3AT, implicating an oxidative cytosolic environment in the PBD cells could be responsible. They then went into whole animals by generating a mouse with a Pex14 DN allele from the wildtype gene, quite similar to a human PBD allele. The homozygous mouse died soon after birth. Neurons of the cerebellum were disordered consistent with a migration defect, as expected; there was more catalase in the cytosol, consistent with a PBD defect, as judged by immunofluorescence. There were mild changes in levels of peroxisome-derived lipids such as plasmalogens in the animals. To study cerebellar defects, a hybrid-strain mouse was constructed that survived a week or two after birth. The strongest anatomical defect in these animals was in Purkinje cells, which had shorter axons and were clearly disrupted in neuronal migration. Swollen axons were more common in neurons isolated in culture from these Pex14 DN animals. Exploring whether the BDNF-TrkB pathway was altered, the authors reported an increase in BDNF in the brain stem (mRNA and protein), an increase in the dominant-negative TrkB-T1 splicing product, and a decrease in the downstream ERK1/2-phosphorylated signal. Overall, the results of this paper suggest increase in BDNF activity and downstream effects in the animals.

This paper is interesting and well written, but there are a number of issues, several of them major:

- (1) Regarding the first section about co-culturing, evidence is suggestive that increased BDNF is the important factor secreted from the glial cells that influence the neurons. The effects are mimicked by addition of recombinant protein. But the critical experiment is lacking: immunodepletion of BDNF from the condition medium, which should attenuate the effect on collateral processes, and then add back BDNF to regain the effect. Without this relationship of BDNF and neuron issues, the results are correlative rather than causal.
- (2) While cytosolic catalase is implicated in the increased secretion of BDNF, evidence is weak! Does inhibition of catalase expression by RNAi attenuate secretion in the culture model?
- (3) Fig. 2B: While panel f is clearly a blow-up of panel d, this is not the clear case for panels e and c. I

can't detect the process at all in panel c. Also, the large dark amorphous blot in panels h and j and not adequately described.

(4) Regarding animals, please state the lost function of the carboxy terminus of Pex14p. (How does the missing carboxy fragment function?) I assume it is known. (Minor point)

(5) Figure 5G: I can observe an increase in nuclear density in the mutant, but not any disorganization. I'm not a neuro-anatomist. Please amplify.

(6) Related to this, why is the peroxisomal phenotype (for example, the increase in VLCFA so mild? Shouldn't all PTS1-related activity be completely blocked? In Fig. 6F-H, a positive control from a more severely affected knock-out mouse (for example, generated by another group) would be helpful to compare phenotypes.

(7) Figure 8C: it would be helpful to have a higher-magnification insert from the control, for comparison.

(8) Please substitute a better blot in Fig. 9E. It is not apparent at all that there is an increase in the T1 allele. Regarding Fig. 9B, from the 9A data, it is not clear how cells are demarcated. This is important for determining TrkB positive dots/cell. I cannot see any borders between cells.

(9) At the end of the Results section, the authors conclude that BDNF-TrkB signaling is NOT involved with the neuronal migration defect of the animals. I was shocked, as I thought that this was the entire point of the manuscript. Please elaborate, as I think I'm missing something really important.

Reviewer #3 (Comments to the Authors (Required)):

This paper proposed a new CNS's cell-to-cell communication in which peroxisome biogenesis in astrocytes affect morphological features of neurons via a BDNF-TrkB pathway in vitro. The authors also newly established PBD model mice harboring a mutation in Pex14, and found that the BDNF-TrkB pathway was altered in this mutant mice (in vivo). Although these two findings obtained by in vitro and in vivo studies are novel and important, the relationship between them (in vitro study and in vivo study) is not clear. There are several issues remaining to prove the main conclusion as detailed below.

Major points:

1. In Figure. 1-3, the authors successfully established an in vitro co-culture system using astrocyte-like cultured cells and primary hippocampal neurons, and found that deficiency of peroxisome biogenesis in the astrocyte-like cell induced abnormal axon development of the neurons (and that is presumably mediated by BDNF secreted by astrocytes). However, whether the same things also happen in vivo is unclear, because of the difference in the cell types (glial cells v.s. neurons, and hippocampus v.s. cerebellum etc) they used in the two situations. Rather, based on the authors results (Fig. 10) and previous studies (Schwartz et al., 1997; Maisonpierre et al., 1990; Rocamora et al., 1993), BDNF from ION neurons but not GFAP+ glial cells in cerebellum seem important to this process. Therefore, I recommend the authors to examine in vitro co-culture system using purkinje cells and the different type of neurons (ION neuron is the best, but a neuronal cell line is also OK) expressing Pex14p-DN, that can bridge the in vitro study and the in vivo study.

Furthermore, I think it would be informative to examine specific deletion of Pex14 or BDNF in astrocytes and then to observe the development of cerebellum in vivo.

2. Although the authors conclude that dysregulation of BDNF-TrkB pathway gives rise to the pathogenesis of cerebellum in PBDs both in the Summary and Abstract, the causal relationship between the dysregulation of BDNF-TrkB and the pathogenesis of cerebellum in PBDs is unclear. I recommend the authors to examine knockdown of inhibitory TrkB-T1 specifically in purkinje cells of Pex14 mutant mice using well-established in utero electroporation system, that can directly show the causal relationship between them.

Minor points:

1. In Fig. 4F, the authors demonstrated that cytosolic catalase is involved in peroxisome deficiency-mediated elevation of BDNF mRNA by treatment with the inhibitor 3AT. I recommend the authors to examine catalase knockdown to directly test the function of catalase in this process.

2. In the title of the Figure 4, the authors used "peroxisomal metabolism". I recommend the authors to use more specific words, such as catalase or something.

3. The signal of TrkB in Fig. 4A and 4D seem ambiguous. I would recommend the authors to obtain higher resolution images.

Reviewer #4 (Comments to the Authors (Required)):

This manuscript explores the molecular mechanisms linking alterations of peroxisome biogenesis to a rather heterogenous group of diseases which includes Zellweger spectrum disorders (ZSD), which all share profound alterations of central nervous system (CNS) development and function. In spite of the ubiquitous nature and distribution of peroxisomes, these pathologies have a marked CNS involvement, hence studying them might provide general principles to approach a much larger pool of genetic and non genetic disorders affecting the nervous system.

The quality of the experiments reported in this manuscript is in general very high. A massive amount of results, which originated from in vivo and ex vivo molecular, cellular and genetic approaches, has been included in fourteen multi-panel figures. These are impressive overall (with the exception of Fig. 5F,G and 9D), demonstrating great commitment of the investigators to address this potentially important scientific problem.

In spite of these positive aspects, the manuscript has been assembled in a manner that does not help the reader to identify the nature of their discoveries or make clear a unifying mechanism explaining why peroxisome deficiency causes alterations in specific types of neurons. Part of the problem is that the authors try to address the many aspects of peroxisome biogenesis disorders in a holistic manner, a daunting task which is probably beyond their immediate reach and not strictly required for publication of their findings. It would be advisable to simplify this manuscript, focussing only on one of the several experimental systems (e.g. co-culture or new genetic model) or brain areas (e.g. cerebellum, hippocampus or cortex). Whilst this decision lies with the authors, I take the liberty to suggest that perhaps the main focus should be the deficits found in the cerebellum, which represent the most advance set of results included in the paper. Accordingly, the discussion should be focussed only on the aspects dealt with in this result section.

I would like also urge the authors to justify better their strategic decision in terms of experimental models (e.g. hippocampal neurons in co-culture experiments), targets (e.g. neurotrophic factors among other secreted proteins) and tissues analysed (e.g. the cerebellum and in particular the Purkinje cells). Whilst their choices are obvious for experts in the field, it might be difficult to understand for other readers. Some rewording might be beneficial (e.g. on page 5 end of "co-culture section"; clarify the meaning of "several peroxisome metabolisms" on page 6; define malformations on page 9).

A few conclusions are not very well supported by the experimental evidence included in the manuscript (e.g. cortex alterations in Fig. 5F) and /or some results are not commented upon (e.g. increased levels of AOX A chain in Fig. 6E). In terms of molecular mechanism, I am puzzled by the hypersensitivity to BDNF found in Purkinje cells from Pex14 Δ C/ Δ C mice. Whilst the overexpression of TrkB T1 could originate from a feedback compensatory mechanism linked to over-secretion of BDNF, it should not lead to an acute toxicity in the presence of high BDNF but to a hypersensitivity to low BDNF concentrations. To this section, I would also add an analysis of full length TrkB phosphorylation, a much more direct and reliable way to monitor TrkB pathway activation, together with AKT and PLC γ phosphorylation analysis.

Replies to the summary of reviewer's comments:

E-1) "As you will see, the reviewers think that the data shown are generally well-performed. However, the reviewers also think that the *in vitro* and *in vivo* assays cannot be compared to each other and do not provide the kind of coherent, valuable study we should publish in *Life Science Alliance*. The fact that the *Pex-14* deficient mice do not present a hippocampal phenotype nor BDNF increase in this brain area reduces the value of the *in vitro* assays using hippocampal neurons.

Reviewer #4 proposes to rather shift the emphasis, concentrating on the cerebellum phenotype only (removal of the hippocampal part altogether), and reviewer #3 agreed during reviewer cross-commenting that this would be a good way to address the discordance of the *in vitro* and *in vivo* results. We agree with this view and can therefore offer to invite submission of a revised version of your work. Importantly, the revised version should be clearly re-focused on the *in vivo* part, and the abstract would need to reflect this shift."

Authors' reply (Au): As Editor and reviewers suggested, we deleted *in vitro* experiment parts including Figures 1-4, Figure S1, and the text addressing these data from the initially submitted manuscript. We here revised the manuscript to re-focus on the cerebellum phenotypes. We also provided new data concerning BDNF upregulation in cultured peroxisome biogenesis-defective neurons to uncover molecular mechanism addressing how dysregulation of BDNF-TrkB signaling causes cerebellar malformation in *Pex14^{ΔC/ΔC}* BL/ICR mouse (Figure 4).

E-2) "More *in vivo* evidence is required (see also reviewer #3), and the description of the phenotype and the origin of the excess BDNF potentially responsible for the phenotype needs to be improved."

Au: Reviewer #3 suggested the knockdown of TrkB-T1 specifically in Purkinje cells of *Pex14^{ΔC/ΔC}* BL/ICR mouse using *in utero* electroporation system. For this experiment, we need to get further permission from the animal ethics committee of Kyushu University to perform additional *in vivo* experiment before starting *in utero* electroporation. For this, it usually takes several months to get mice showing the knockdown of TrkB-T1 specifically in Purkinje cells of *Pex14^{ΔC/ΔC}* BL/ICR mouse. Therefore, alternatively, we attempted to specifically knockdown TrkB-T1 in primary culture of cerebellar neurons by *in vitro* electroporation of short hairpin RNA for TrkB-T1 to investigate the causal relationship between dysregulation of BDNF-TrkB signaling and pathogenesis of cerebellum in *Pex14^{ΔC/ΔC}* BL/ICR mouse. We optimized the electroporation condition for knocking down of TrkB-T1 in Purkinje cells. However, unfortunately, analysis of Purkinje cell morphology was failed because of detachment of Purkinje cells at 10 days *in vitro* (DIV), four days before of the morphological analysis at 14 DIV such as those shown in Figure 5C and G. Instead, we provide several additional *in vivo* data concerning impaired BDNF-TrkB signaling in cerebellum of *Pex14^{ΔC/ΔC}* BL/ICR mouse, including a reduced level of phosphorylation of TrkB-TK+ and AKT (Figure 7A, B, and D). Moreover, the BDNF level was elevated in the brain stem region of *Pex14^{ΔC/ΔC}* BL/ICR mouse (Figure 8H). These results are consistent with the previous studies on BDNF- and TrkB variant-mediated neuronal morphogenesis (Yacoubian and Lo, 2000, cited reference), where the elevated BDNF and TrkB-T1 inhibit the phosphorylation of TrkB-TK+, resulting in the neuronal morphological alteration. Therefore, it is most likely that impairment of BDNF-TrkB signaling pathway causes malformation of Purkinje cells.

We added detailed descriptions of the phenotypes in *Pex14^{ΔC/ΔC}* mouse including cortical layer structure (Figure 1F and G), peroxisomal matrix protein import (Figure 2E), and affected peroxisomal metabolism (Figure 2F-H). We also provided better-quality Figure 6A and D.

To investigate the origin of an elevated level of BDNF potentially responsible for the dysmorphogenesis of Purkinje cells, we tried to analyze the morphology of Purkinje cells by co-culturing with primary brain stem neurons. However, primary brain stem neurons did not survive up to the day of morphological analysis of Purkinje cells, namely 14 days of the

culture *in vitro*. Alternatively, organ culture system might be useful to address this issue, although this experiment can be done using organs such as rat cerebellar slice at postnatal day 9 and brain stem explant containing inferior olive at embryonic day 15 (Uesaka et al., 2012). Accordingly, more sophisticated experimental conditions seem to be required for the co-culture experiment of Purkinje cells with neurons derived from brain stem.

Instead, we rather provided the data showing upregulated protein level of BDNF in brain stem region of *Pex14^{AC/AC}* BL/ICR mouse (Figure 8H). This additional *in vivo* evidence together with *in vitro* data showing dysmorphogenesis of Purkinje cells in the presence of excess BDNF (Figure 5C-H) lead us to conclude that peroxisome deficiency elevates the expression of BDNF in ION and TrkB-T1 in cerebellum that attenuates BDNF–TrkB signaling, giving rise to the malformation of cerebellum.

Reference (cited here in this reply, but not in the manuscript)

Uesaka N., T. Mikuni, K. Hashimoto, H. Hirai, K. Sakimura, M. Kano. 2012. Organotypic coculture preparation for the study of developmental synapse elimination in mammalian brain. *J. Neurosci.* 32:11657-11670.

Reviewer #1

We thank the Reviewer #1 very much for the helpful comments and suggestions.

Editor and other reviewers suggested us that manuscript should be concentrated only on the cerebellum phenotype. Therefore, we deleted *in vitro* experiment parts including Figures 1-4, Figure S1, and the text addressing these data from the initially submitted manuscript. We here revised the manuscript to re-focus on the cerebellum phenotypes. We also provided new data concerning BDNF upregulation in cultured peroxisome biogenesis-defective neurons to uncover molecular mechanism addressing how dysregulation of BDNF-TrkB signaling causes cerebellar malformation in *Pex14^{ΔC/ΔC}* BL/ICR mouse (Figure 4). In this study, *Pex14^{ΔC/ΔC}* mice on a C57BL/6 background die within several days after birth, and thereby we can analyze only the cortical phenotype at P0.5. *Pex14^{ΔC/ΔC}* BL/ICR mice generated by the mating of *Pex14^{ΔC/ΔC}* mice with ICR background mouse survive longer time than *Pex14^{ΔC/ΔC}* mice and were used for the analysis of cerebellar phenotype. Revised parts were marked in red in the text.

Replies to the comments:

R1-1) "... *But the critical experiment is lacking: immunodepletion of BDNF from the condition medium, which should attenuate the effect on collateral processes, and then add back BDNF to regain the effect. Without this relationship of BDNF and neuron issues, the results are correlative rather than causal.*"

Au: As describe above, we revised the manuscript by taking away the *in vitro* data concerning primary hippocampus neurons from the initially submitted version of manuscript. Thank you for the helpful suggestion.

R1-2) "*Cytosolic catalase is implicated in the increased secretion of BDNF, evidence is weak! Does inhibition of catalase expression by RNAi attenuate secretion in the culture model?*"

Au: In the revised manuscript, we excluded the data concerning involvement of cytosolic catalase in the increased secretion of BDNF from peroxisome-deficient RCR-1 cells. Thank you for your critical comments.

R1-3) "*Fig 2B: While panel f is blow-up of panel d, this is not the clear case for panels e and c. I can't detect the process at all in panel c. Also, the large dark amorphous blot in panels h and j and not adequately described.*"

Au: We excluded Figure 2B in the revised manuscript. Thank you for your helpful and critical comments.

R1-4) "*Regarding animals, please state the lost function of the carboxyl terminus of Pex14p. (How does the missing carboxyl fragment function?) I assume it is known. (Minor point)*"

Au: The C-terminal region of Pex14p contains coiled-coil domain (encompassing the amino acid residues of positions, 157-197, Figure 1C), is involved in the homo-oligomerization for the effective peroxisomal protein import. We added the possible mechanism regarding to the affected peroxisomal matrix protein import in *Pex14^{ΔC/ΔC}* mouse in the section of Discussion (Page 10, lines 7-25).

R1-5) "*Figure 5G: I can observe an increase in nuclear density in the mutant, but not any disorganization. I'm not a neuro-anatomist. Please amplify.*"

Au: In the initially submitted manuscript, explanation of disorganization of cortical neurons was not sufficient. Therefore, we clearly described the neurological phenotype in cortex of *Pex14^{ΔC/ΔC}* mouse in the Results section (Page 4, lines 18-22).

R1-6) "*Related to this, why is the peroxisomal phenotype (for example, the increase in VLCFA) so mild? Shouldn't all PTS1-related activity be completely blocked? In Fig 6F-H, a positive control from a more severely affected knock-out mouse (for example, generated by another group) would be helpful to compare phenotypes.*"

Au: Peroxisomal import of matrix proteins including PTS1 and PTS2 enzymes is partially affected in *Pex14^{AC/AC}* mouse, as shown in immunofluorescent microscopy (Figure 2A and B in the revised manuscript) and AOx processing (Figure 2E in the revised manuscript). Only PTS1-like catalase import is completely impaired in this mouse (Figure 2C in the revised manuscript). These results suggest that N-terminal region of Pex14p could partially translocate PTS1 and PTS2 proteins, but not catalase, into peroxisomes. In Discussion, we described the function of coiled-coil domain of Pex14p in peroxisome matrix protein import. In regard to metabolic abnormalities in *Pex14^{AC/AC}* mouse, the results were not sufficiently described in the initially submitted manuscript. We, therefore, revised the manuscript by comparing our *Pex14^{AC/AC}* mouse to other *Pex* knockout mice, including *Pex2*⁻, *Pex5*⁻, and *Pex13*-knockout mice (from page 4, line 36 to page 5, line 9).

R1-7) “*Figure 8C: it would be helpful to have a higher-magnification insert from the control, for comparison.*”

Au: We thank you for helpful suggestion. Enlarged views of axonal morphologies in control cells and non-treated *Pex14*-deficient cells are shown in the insets of revised Figure 5C in the revised manuscript.

R1-8) “*Please substitute a better blot in Fig. 9E. It is not apparent at all that there is an increase in the T1 allele. Regarding Fig. 9B, from the 9A data, it is not clear how cells are demarcated. This is important for determining TrkB-positive dots/cell. I cannot see any borders between cells.*”

Au: We revised the immunoblot data in Figure 6E (Figure 9E in the initially submitted manuscript) showing an elevated level of TrkB-T1 in *Pex14^{AC/AC}* BL/ICR mouse. In Figure 6A (Figure 9A in the initially submitted manuscript), cell surfaces were indicated by dashed lines to discriminate the cell borders.

R1-9) “*At the end of the Results section, the authors conclude that BDNF-TrkB signaling is NOT involved with the neuronal migration defect of the animals. I was shocked, as I thought that this was the entire point of the manuscript. Please elaborate, as I think I'm missing something really important.*”

Au: Since neuronal migration defect is a characteristic phenotype in *Pex*-knockout mice and patients with peroxisome biogenesis disorders, the Reviewer #1's interest might likely be the mechanisms underlying neuronal migration defect. This manuscript provided several data reporting that peroxisomal biogenesis deficiency causes the impairment of BDNF-TrkB signaling in Purkinje cells, giving rise to the malformation of Purkinje cells in the cerebellum.

In the peroxisome-deficient mice including *Pex14^{AC/AC}* mouse and other *Pex*-knockout mice, migration defects of projection neurons are observed in the cortex (Figure 1F and G in the revised manuscript) Protein levels of BDNF and TrkB variants (Figure S4A-C in the revised manuscript) and phosphorylation level of ERK1/2 and AKT were not altered in cortex (Figure S4D and E in the revised manuscript). Collectively, these data suggest that neuronal migration defect in cortex is unlikely caused by an impaired BDNF-TrkB signaling as described in the last paragraph of the Results section (Page 9, lines 20-29).

Reviewer #3

We thank you very much for the helpful comments and suggestions.

Editor, other reviewer, and you suggested us that manuscript should be concentrated only on the cerebellum phenotype. Therefore, we deleted *in vitro* experiment parts including Figures 1-4, Figure S1, and the text addressing these data from the initially submitted manuscript. We here revised the manuscript to re-focus on the cerebellum phenotypes. We also provided new data concerning BDNF upregulation in cultured peroxisome biogenesis-defective neurons to uncover molecular mechanism addressing how dysregulation of BDNF-TrkB signaling causes cerebellar malformation in *Pex14^{AC/AC}* BL/ICR mouse (Figure 4). In this study, *Pex14^{AC/AC}* mice on a C57BL/6 background die within several days after birth, and thereby we can analyze only the cortical phenotype at P0.5. *Pex14^{AC/AC}* BL/ICR mice generated by the mating of *Pex14^{AC/AC}* mice with ICR background mouse survive longer time than *Pex14^{AC/AC}* mice and were used for the analysis of cerebellar phenotype. Revised parts were marked in red in the text.

Replies to the comments:

R3-1) “...However, whether the same things also happen *in vivo* is unclear, because of the difference in the cell types (glial cells vs. neurons, and hippocampus vs. cerebellum etc.) they used in the two situations. Rather, based on the authors results (Fig 10) and previous studies (...), BDNF from ION neurons but not GFAP+ glial cells in cerebellum seem important to this process. Therefore, I recommend the authors to examine *in vitro* co-culture system using Purkinje cells and the different types of neurons (ION neuron is the best, but a neuronal cell line is also OK) expressing *Pex14p-DN*, that can bridge the *in vitro* study and the *in vivo* study.”

Authors' reply (Au): To investigate the origin of an elevated level of BDNF potentially responsible for the dysmorphogenesis of Purkinje cells, we tried to analyze the morphology of Purkinje cells by co-culturing with primary brain stem neurons. However, primary brain stem neurons did not survive up to the day of morphological analysis of Purkinje cells, namely 14 days of the culture *in vitro*. Alternatively, organ culture system might be useful to address this issue, although this experiment can be done using organs such as rat cerebellar slice at postnatal day 9 and brain stem explant containing inferior olive at embryonic day 15 (Uesaka et al., 2012). Accordingly, more sophisticated experimental conditions seem to be required for the co-culture experiment of Purkinje cells with neurons derived from brain stem.

Instead, we rather provided the data showing upregulated protein level of BDNF in brain stem region of *Pex14^{AC/AC}* BL/ICR mouse (Figure 8H). This additional *in vivo* evidence together with *in vitro* data showing dysmorphogenesis of Purkinje cells in the presence of excess BDNF (Figure 5C-H) lead us to conclude that peroxisome deficiency elevates the expression of BDNF in ION and TrkB-T1 in cerebellum that attenuates BDNF-TrkB signaling, giving rise to the malformation of cerebellum.

R3-2) “Furthermore, I think it would be informative to examine specific deletion of *Pex14* or *BDNF* in astrocytes and then to observe the development of cerebellum *in vivo*.”

Au: Thank you for your helpful and critical comments. In the initially submitted manuscript, we showed that the elevated BDNF level in cerebellum of *Pex14^{AC/AC}* BL/ICR mouse is less likely caused by the increased expression of BDNF in Bergman glia and Purkinje cells. From these observations, we suggested that ION neurons play a role in the elevation of BDNF in cerebellum of *Pex14^{AC/AC}* BL/ICR mouse. This conclusion is further supported by the additional *in vivo* data showing the upregulated protein level of BDNF in brain stem region of *Pex14^{AC/AC}* BL/ICR mouse (Figure 8H). Moreover, we show that elevation of BDNF transcription is caused by the impaired peroxisome biogenesis in SH-SY5Y (Fig 4). Collectively, these results in the revised manuscript suggest that dysregulation of BDNF transcription in ION neurons, rather than astrocyte most likely causes malformation of Purkinje cells via dysregulation of BDNF-TrkB signaling during the postnatal development. Therefore, we think that generation of mouse specifically deleted *Pex14* or *BDNF* in astrocyte may not be required.

R3-3) “Although the authors conclude that dysregulation of BDNF-TrkB pathway gives rise to the pathogenesis of cerebellum in PBDs both in the Summary and Abstract, the causal relationship between the dysregulation of BDNF-TrkB and the pathogenesis of cerebellum in PBDs is unclear. I recommend the authors to examine knockdown of inhibitory TrkB-T1 specifically in Purkinje cells of *Pex14* mutant mice using well-established in utero electroporation system, that can directly show the causal relationship between them.”

Au: The reviewer #3 suggested the knockdown of TrkB-T1 specifically in Purkinje cells of *Pex14*^{ΔC/ΔC} BL/ICR mouse using in utero electroporation system. For this experiment, we need to get further permission from the animal ethics committee of Kyushu University to perform additional *in vivo* experiment before starting in utero electroporation. For this, it usually takes several months to get mice showing the knockdown of TrkB-T1 specifically in Purkinje cells of *Pex14*^{ΔC/ΔC} BL/ICR mouse. Therefore, alternatively, we attempted to specifically knockdown TrkB-T1 in primary culture of cerebellar neurons by *in vitro* electroporation of short hairpin RNA for TrkB-T1 to investigate the causal relationship between dysregulation of BDNF-TrkB signaling and pathogenesis of cerebellum in *Pex14*^{ΔC/ΔC} BL/ICR mouse. We optimized the electroporation condition for knocking down of TrkB-T1 in Purkinje cells. However, unfortunately, analysis of Purkinje cell morphology was failed because of detachment of Purkinje cells at 10 days in vitro (DIV), four days before of the morphological analysis at 14 DIV such as those shown in Figure 5C and G. Instead, we provide several additional *in vivo* data concerning impaired BDNF-TrkB signaling in cerebellum of *Pex14*^{ΔC/ΔC} BL/ICR mouse, including a reduced level of phosphorylation of TrkB-TK+ and AKT (Figure 7A, B, and D). Moreover, the BDNF level was elevated in the brain stem region of *Pex14*^{ΔC/ΔC} BL/ICR mouse (Figure 8H). These results are consistent with the previous studies on BDNF- and TrkB variant-mediated neuronal morphogenesis (Yacobian and Lo, 2000, cited reference), where the elevated BDNF and TrkB-T1 inhibit the phosphorylation of TrkB-TK+, giving rise to the neuronal morphological alteration. Therefore, it is most likely that impairment of BDNF-TrkB signaling pathway causes malformation of Purkinje cells.

Minor points

R3-4) “In Fig. 4F, the authors demonstrated that cytosolic catalase is involved in peroxisome deficiency-mediated elevation of BDNF mRNA by treatment with the inhibitor 3AT. I recommend the authors to examine catalase knockdown to directly test the function of catalase in this process.”

Au: In the revised manuscript, we excluded the data concerning involvement of cytosolic catalase in the increased secretion of BDNF from peroxisome-deficient RCR-1 cells, as suggested by the editors and Reviewers. Thank you for your suggestions.

R3-5) “In the title the Figure 5, the authors used “peroxisomal metabolism”. I recommend the authors to use more specific words, such as catalase or something.”

Au: “peroxisomal metabolism” was used in the title of Figure “4” of the initially submitted manuscript. We excluded Figure 4 in the revised manuscript. We thank you for the helpful suggestions.

R3-6) “The signal of TrkB in Fig. 4A and 4D seem ambiguous. I would recommend the authors to obtain higher resolution images.”

Au: This Reviewer #3 seems to suggest us to show the higher resolution image of Figure 9A and D in the initially submitted manuscript. Higher magnification images were provided in the Figure 6A and D in the revised manuscript. In addition, cell surfaces were indicated by dashed lines to discriminate cell borders in the Figure 6A.

Reference (cited here in this reply, but not in the manuscript)

Uesaka N., T. Mikuni, K. Hashimoto, H. Hirai, K. Sakimura, M. Kano. 2012. Organotypic coculture preparation for the study of developmental synapse elimination in mammalian brain. *J. Neurosci.* 32:11657-11670.

Reviewer #4

We thank the Reviewer #4 very much for the helpful comments and suggestions.

Editor, other reviewer, and you suggested us that manuscript should be concentrated only on the cerebellum phenotype. Therefore, we deleted *in vitro* experiment parts including Figures 1-4, Figure S1, and the text addressing these data from the initially submitted manuscript. We here revised the manuscript to re-focus on the cerebellum phenotypes. We also provided new data concerning BDNF upregulation in cultured peroxisome biogenesis-defective neurons to uncover molecular mechanism addressing how dysregulation of BDNF-TrkB signaling causes cerebellar malformation in *Pex14^{ΔC/ΔC}* BL/ICR mouse (Figure 4). In this study, *Pex14^{ΔC/ΔC}* mice on a C57BL/6 background die within several days after birth, and thereby we can analyze only the cortical phenotype at P0.5. *Pex14^{ΔC/ΔC}* BL/ICR mice generated by the mating of *Pex14^{ΔC/ΔC}* mice with ICR background mouse survive longer time than *Pex14^{ΔC/ΔC}* mice and were used for the analysis of cerebellar phenotype. Revised parts were marked in red in the text.

Replies to the comments:

R4-1) “*These are impressive overall (with exception of Fig. 5F,G and 9D)...*”

Authors’ reply (Au): In the initially submitted manuscript of Figure 5G, explanation of disorganization of cortical neurons was not sufficient. Therefore, we clearly described the neurological phenotype in cortex of *Pex14^{ΔC/ΔC}* mouse in the Results section (Page 4, lines 18-22). In addition, we added arrows indicating accumulated neurons in Figure 1F (Figure 5 F in the initially submitted manuscript). In the Figure 6D (Figure 9D in the initially submitted manuscript), higher-magnification images were shown.

R4-2) “*A few conclusions are not very well supported by the experimental evidence included in the manuscript (e.g. cortex alterations in Fig 5F) and /or some results are not commented upon (e.g. increased levels of AOX A chain in Fig 6E).*”

Au: As described in R4-1, we revised the description of neurological phenotype in cortex of *Pex14^{ΔC/ΔC}* mouse (Figure 1G in the revised manuscript) in the Results section (Page 4, lines 18-22). In addition, we added the explanation of unprocessed AOX A-chain in *Pex14^{ΔC/ΔC}* mouse (Figure 2E; 6E in the initially submitted manuscript) in Results section (Page 4, lines 29-30). Furthermore, we clearly described the metabolic abnormalities in *Pex14^{ΔC/ΔC}* mouse by comparing to other generalized *Pex* knockout mice in the revised manuscript (from Page 4, line 36 to page 5, line 9).

R4-3) “*In terms of molecular mechanism, I am puzzled by the hypersensitivity to BDNF found in Purkinje cells from Pex14 deltaC/deltaC mice. Whilst the overexpression of TrkB T1 could originate from a feedback compensatory mechanism linked to over-secretion of BDNF, it should not lead to an acute toxicity in the presence of high BDNF but not a hypersensitivity to low BDNF concentrations.*”

Au: As shown in Figure S3C and D in the revised manuscript, BDNF did not enhance the TrkB-T1 expression in the primary cerebellar neuron. Therefore, feedback upregulation of TrkB-T1 does not appear to take place by an elevated level of BDNF. We added the explanation of this experiments in the Results section (Page 8, lines 25-27).

On the other hand, the Reviewer #4 suggests that TrkB-T1-elevated cells show the hypersensitivity to BDNF, but not the acute toxicity in the presence of high BDNF. As

described in Discussion (Page 12, lines 3-5), Yacoubian and Lo reported that excess BDNF inhibits dendritic growth of the cells expressing TrkB-T1 by inhibiting TrkB-TK+ activation (Yacoubian and Lo, 2000, cited reference). Indeed, phosphorylation level of TrkB-TK+ was decreased in cerebellum (Figure 7A and B in the revised manuscript). Therefore, upregulated TrkB-T1 in Purkinje cells most likely shows inhibitory effect on TrkB-TK+ signaling, leading to the impairment of Purkinje cells development.

R4-4) “*To this section, I would also add an analysis of full length TrkB phosphorylation, a much more direct and reliable way to monitor TrkB pathway activation, together with AKT and PLCgamma phosphorylation analysis.*”

Au: The initial step of BDNF-TrkB signaling pathway is the autophosphorylation of cytosolic tyrosine kinase domain of TrkB-TK+. Decreased level of phosphorylated TrkB-TK+ was observed in the cerebellum of *Pex14^{ΔC/ΔC}* BL/ICR mouse (Figure 7A and B; Figure S3 H in the revised manuscript). We also analyzed phosphorylated level of AKT and PLCγ in the cerebellum. Phosphorylated AKT was decreased in *Pex14^{ΔC/ΔC}* BL/ICR mouse, while phosphorylated PLCγ was not altered (Figure 7A, D, and E in the revised manuscript). Moreover, as shown in Figure S3E in the revised manuscript, both *c-fos* and *c-jun* mRNAs, target gene of BDNF-TrkB signaling pathway, were decreased in the *Pex14^{ΔC/ΔC}* BL/ICR mouse cerebellum, indicating the deactivation of BDNF-TrkB signaling pathway.

November 5, 2018

RE: Life Science Alliance Manuscript #LSA-2018-00062-TR

Prof. Yukio Fujiki
Kyushu University
International Institute for Carbon-Neutral Energy Research
744 Motooka, Nishi-ku
Higashi-ku
Fukuoka 819-0395
Japan

Dear Dr. Fujiki,

Thank you for submitting your revised manuscript entitled "Peroxisome biogenesis deficiency attenuates the BDNF-TrkB pathway-mediated development of cerebellum".

The original reviewers evaluated your manuscript again and they all appreciate the introduced changes. We would be thus happy to publish your paper in Life Science Alliance pending final revisions necessary to address a few remaining concerns of reviewer #1. No new experiments are needed, but please amend your manuscript further following the suggestions made by this reviewer. Please also provide your ORCID iD, you should have received an email with instructions on how to do so.

A. FINAL FILES:

-- High-resolution figure, supplementary figure and video files uploaded as individual files: See our detailed guidelines for preparing your production-ready images, <http://life-science-alliance.org/authorguide>

B. MANUSCRIPT ORGANIZATION AND FORMATTING:

Full guidelines are available on our Instructions for Authors page, <http://life-science-alliance.org/authorguide>

Sincerely,

Reviewer #1 (Comments to the Authors (Required)):

The work of Abe et al. provides a novel mechanism to explain cerebellar defects in patients with peroxisomal assembly defects such as Zellweger Spectrum Disorder. A mouse has been

constructed with a Pex14 mutation such that domains toward the carboxy end are deleted. Neuronal migration defects in the embryo are observed. A second mouse model homozygous for the Pex14 deletion but in a different background yields surviving neonates which can be analyzed more fully. An increase in BDNF and downstream signaling is observed, which leads to morphological and developmental neurological development. These changes may be tied to a change in cytoplasmic redox state, as catalase fails to be imported into peroxisomes.

This is a revision that focuses more on the mice and omits a previous large section on in vitro effects. I had problems relating this section to the rest of the paper, and this section has now been deleted. The authors also have adequately addressed by prior concerns. Overall, the paper holds together much better. While I still have a few minor issues (see below), I think the work significantly adds to our knowledge of how defects of peroxisomal assembly lead to severe neurological sequelae in very young human patients.

Remaining issues:

(1) High magnification insets are needed in panels A-D of Figure 2 (such as those provided in Fig. 4B). They can be placed in Supplementary if necessary.

(2) For clarity, when p75^{ECD}-His is introduced on page 6, state that this domain competes for ligand with the intact receptor.

(3) One gets the impression early on that an increase in BDNF alone may mediate the effects of the Pex14 truncation on neuronal development. It is somewhat confusing then that the neuronal effects depend BOTH on the truncation AND increased exogenous BDNF, not on an increase in BDNF alone. It this should be made clearer in the description/discussion of Fig. 5F. So the Pex14 truncation must have other effects on neuronal development (if I understand the experiment correctly).

(4) In Fig 4F, BDNF causes axon swelling, while in Fig. 4H BDNF causes a decrease in cell area. Are these two opposing effects related? Perhaps some comment in the text would help.

Reviewer #3 (Comments to the Authors (Required)):

The revised manuscript improved significantly, and the authors have replied to my previous comments and questions appropriately. I am satisfied with the reply by authors and new data of the revised manuscript. The data is potentially interesting and worthy of publication.

Reviewer #4 (Comments to the Authors (Required)):

This revised manuscript (duly amended following the suggestions of editor and reviewers) is much more focussed and easier to read. Whilst some key experiments are still missing, this is ultimately due to technical problems rather than lack of willingness in boosting the scientific weight of the manuscript. Altogether I am positive about this amended manuscript and I have no further comments, besides encouraging the authors to persevere in the attempt to deliver in utero electroporation to validate the overall model in a future manuscript.

2nd Authors' Response to Reviewers: November 8, 2018

Reply to the Reviewer's comment

Reviewer #1

We thank the Reviewer #1 very much for the helpful comments and suggestions.

1) "High magnification insets are needed in panels A-D of Figure 2 (such as those provided in Fig. 4B). They can be placed in Supplementary if necessary."

Authors' reply (Au): We thank you for helpful suggestion. Enlarged views of peroxisomal morphologies in cortex of *Pex14^{ΔC/ΔC}* mouse are shown in the insets of the revised Figure 2A-D.

2) "For clarity, when p75ECD-His is introduced on page 6, state that this domain competes for ligand with the intact receptor."

Au: In the revised manuscript, we added the sentence concerning the function of p75ECD-His in the neurite outgrowth of *PEX5*-knocked down SH-SY5Y cells (page 6, line 20).

3) "One gets the impression early on that an increase in BDNF alone may mediate the effects of the Pex14 truncation on neuronal development. It is somewhat confusing then that the neuronal effects depend BOTH on the truncation AND increased exogenous BDNF, not on an increase in BDNF alone. It this should be made clearer in the description/discussion of Fig. 5F. So the Pex14 truncation must have other effects on neuronal development (if I understand the experiment correctly)."

Au: Truncation of the C-terminal half part of Pex14p causes peroxisome deficiency (Figure 2). Therefore, we modified the text to clearly describe that peroxisome deficiency leads to the impairment of BDNF-induced morphogenesis in primary culture of Purkinje cells (page 7, lines 19-20) and the dysfunction of BDNF-TrkB signaling pathway caused by the combination of elevated BDNF and upregulation of TrkB-T1 in *Pex14^{ΔC/ΔC}* BL/ICR mouse (page 9, lines 4-6).

4) "In Fig 4F, BDNF causes axon swelling, while in Fig. 4H BDNF cause a decrease in cell area. Are these opposing effects related? Perhaps some comment in the text would help."

Au: As a result of primary culture of cerebellar neurons, both the defect of dendritic development and the axonal swelling in Purkinje cells of *Pex14^{ΔC/ΔC}* BL/ICR mouse are caused by elevated levels of BDNF and TrkB-T1 (Fig 5C-H). Defect of dendritic development is most likely owing to the dominant-negative effect of TrkB-T1 on TrkB-TK+ signaling, since the defect of dendritic arborization is consistent with the report by Yacoubian & Lo (Yacoubian &

Lo, 2000), where they report that excess BDNF inhibits the dendritic growth of the neurons expressing TrkB-T1. However, the downstream of BDNF signaling pathway in the axonal swelling remains to be defined. As Reviewer #1 suggested, we therefore revised the sentence addressing the dominant-negative effect of TrkB-T1 on TrkB-TK+ signaling in the defect of dendritic arborization and modified the text describing the relationship of signaling between axonal swelling and the defect of dendritic arborization (page 12, lines 3-8). Further study is needed to elucidate the downstream of elevated levels of BDNF and TrkB-T1 that leads to the axonal swelling.

Reviewer #3

We thank the Reviewer #3 very much for encouraging comments

Reviewer #4

We thank the Reviewer #4 very much for kind comments and encouragement.

2nd Revision - Editorial Decision November 8, 2018

RE: Life Science Alliance Manuscript #LSA-2018-00062-TRR

Prof. Yukio Fujiki
Kyushu University
Medical Institute of Bioregulation
3-1-1 Maidashi
Higashi-ku
Fukuoka, Fukuoka 812-8582
Japan

Dear Dr. Fujiki,

Thank you for submitting your Research Article entitled "Peroxisome biogenesis deficiency attenuates the BDNF-TrkB pathway-mediated development of cerebellum". We appreciate the introduced changes, and it is a pleasure to let you know that your manuscript is now accepted for publication in Life Science Alliance. Congratulations on this interesting work.

DISTRIBUTION OF MATERIALS:

Again, congratulations on a very nice paper. I hope you found the review process to be constructive and are pleased with how the manuscript was handled editorially. We look forward to future exciting submissions from your lab.

Sincerely,

Andrea Leibfried, PhD
Executive Editor
Life Science Alliance
Meyerhofstr. 1

69117 Heidelberg, Germany
t +49 6221 8891 502
e a.leibfried@life-science-alliance.org
www.life-science-alliance.org